# Rethinking simultaneous suppression in visual cortex via compressive spatiotemporal population receptive fields

Eline R. Kupers [1] ✉, Insub Kim [1] & Kalanit Grill-Spector [1,2]

When multiple visual stimuli are presented simultaneously in the receptive field, the neural response is suppressed compared to presenting the same stimuli sequentially. The prevailing hypothesis suggests that this suppression is due to competition among multiple stimuli for limited resources within receptive fields, governed by task demands. However, it is unknown how stimulus-driven computations may give rise to simultaneous suppression. Using fMRI, we find simultaneous suppression in single voxels, which varies with both stimulus size and timing, and progressively increases up the visual hierarchy. Using population receptive field (pRF) models, we find that compressive spatiotemporal summation rather than compressive spatial summation predicts simultaneous suppression, and that increased simultaneous suppression is linked to larger pRF sizes and stronger compressive nonlinearities. These results necessitate a rethinking of simultaneous suppression as the outcome of stimulus-driven compressive spatiotemporal computations within pRFs, and open new opportunities to study visual processing capacity across space and time.

The human visual system has limited processing capacity. We are worse at processing multiple stimuli presented at once than when the identical stimuli are shown one after the other in the same location. This drop in performance has been observed in a variety of visual tasks, such as searching for a target among distractors[1,2], recognizing an object when surrounded by flankers[3], or keeping multiple items in short-term visual working memory[4].

A neural phenomenon attributed to limited visual capacity is simultaneous suppression: a reduced response when multiple visual stimuli are presented at once than when the identical stimuli are shown one after the other in sequence[5–12]. Simultaneous suppression is prevalent and robust: it is observed across the visual cortex, from the level of single-neuron spiking[5–7], all the way to the level of entire visual areas using functional magnetic resonance imaging (fMRI)[8–12], with large effect sizes: up to two-fold amplitude differences between sequential and simultaneous presentations of otherwise identical stimuli[8,9,12].

A prevailing explanation linking simultaneous suppression to visual capacity is based on the influential theory of biased competition[7,8,13]. This theory argues that visual processing capacity is determined by the computational resources afforded by receptive fields, where the visual system prioritizes inputs that are behaviorally relevant for further processing. When a visual stimulus is presented alone in the receptive field, the item can be fully processed with limited neural resources. However, when multiple stimuli are presented at once in the receptive field, the stimuli are hypothesized to compete for neural resources, resulting in a reduced neural response. This explanation of stimuli competing for neural resources is linked to visual attention, as it has been suggested that competition can be governed by task or behavioral demands[13]. As such, a large body of research has examined how simultaneous suppression is modulated by visual attention[7,8,14–16] and stimulus context[10,11]. However, it is unknown how stimulus-driven computations within receptive fields may give rise to simultaneous suppression in the first place. Thus, the goal of the

[1]Department of Psychology, Stanford University, Stanford, CA, USA. [2]Wu Tsai Neurosciences Institute, Stanford University, Stanford, CA, USA.
✉e-mail: ekupers@stanford.edu

present study is to operationalize and elucidate the computational mechanisms underlying simultaneous suppression in the human visual cortex.

A key prediction stemming from the biased competition theory is that simultaneous suppression will only occur in neurons whose receptive fields are large enough to encompass several stimuli[9,12]. It is well documented that the size of receptive fields[17] and population receptive fields (pRFs, aggregate receptive field of the neuronal population in an fMRI voxel[18,19]) progressively increase from earlier to higher areas up the visual hierarchy. Consistent with this idea, several studies reported that simultaneous suppression systematically increases up the visual hierarchy and is absent in V1[7–9,12], suggesting that the lack of suppression in V1 is because its receptive fields are too small to encompass multiple visual stimuli. However, there is an assumption in prior work, which is that neurons sum inputs linearly over the duration of the stimulus[7,9,12,20]. This assumption has emphasized research on how the spatial overlap between stimuli and the receptive field may affect simultaneous suppression[8,9,12] and less on how stimulus timing may affect simultaneous suppression.

As stimuli have identical duration, size, and location across simultaneous and sequential conditions, neurons summing linearly over visual space and linearly over stimulus duration are predicted to respond identically in these conditions. Therefore, the lower responses for simultaneous than sequential presentations suggest nonlinear, and in particular, subadditive summation. Although linear summation within receptive fields is observed in some cases[21–23], violations of response linearity in the human visual system have been extensively reported. Spatially, responses to bigger stimuli are typically less than the sum of responses to smaller stimuli[22,24–27]. Temporally, responses to longer stimuli are typically smaller than the sum of responses to shorter stimuli[28–44]. While both hemodynamic[45–47] and neural[27,35,39–42,48–51] mechanisms may contribute to observations made with fMRI, empirical and computational modeling studies suggest that the observed subadditivity is driven by compressive summation of visual inputs within neurons' receptive fields across space[27,50] and across time[33,36,38,40–44]. Therefore, rather than considering how context or task demands affect simultaneous suppression, we asked: To what extent is simultaneous suppression a consequence of compressive computations within visual receptive fields? Here, we consider two possible classes of compressive neural mechanisms that may generate simultaneous suppression: compressive spatial summation (CSS) and compressive spatiotemporal (CST) summation.

CSS within the receptive field predicts that the response to multiple stimuli presented together within the pRF (as in simultaneous conditions) will be lower than the sum of responses to the individual stimuli shown alone (as in sequential conditions). As the duration of stimuli is matched between the simultaneous and sequential conditions, the spatial summation hypothesis predicts that the level of simultaneous suppression will depend only on the spatial overlap between the stimuli and the pRF. As both average receptive field size and compressive nonlinearities increase up the visual hierarchy[27,50], CSS also predicts that the level of simultaneous suppression will increase from earlier to later visual areas.

CST summation within the receptive field predicts that simultaneous suppression will not only depend on the spatial overlap between the stimuli and the receptive field, but also on the timing of stimuli. This prediction is based on the empirical observation that neuronal responses to visual stimuli vary over time, typically showing an initial strong transient response at stimulus onset (lasting for 100–200 ms), followed by a weaker sustained response lasting for the duration of the stimulus[32,42,44,52–55], and often a transient response at stimulus offset[42,44,54]. These nonlinear temporal dynamics suggest that presenting all stimuli at once in the pRF (as in simultaneous conditions) results in two transients (at stimulus onset and offset). This response will be lower than the accumulated response induced by multiple visual transients when presenting multiple stimuli one by one in a rapid fashion (as in sequential conditions). Thus, the spatiotemporal hypothesis predicts that the level of simultaneous suppression will depend both on the spatial overlap between the stimuli and the pRF, and the difference in the number of visual transients between simultaneous and sequential conditions. Similar to compressive spatial nonlinearities, compressive temporal nonlinearities also increase the visual hierarchy[39–42,44,51], predicting an increase in the level of suppression as pRF size and spatiotemporal compression increase.

Here, we used fMRI and a computational pRF framework to distinguish between these hypotheses. We conducted two fMRI experiments. In the first (SEQ-SIM, Fig. 1a), we measured responses to sequentially or simultaneously presented stimuli and examined how stimulus size and timing affect the level of simultaneous suppression in each voxel of the visual system (Fig. 1b). In the second experiment (retinotopy, Fig. 1d), we estimated each voxel's spatial pRF parameters and used estimated parameters in a pRF modeling framework to predict the blood oxygen level-dependent (BOLD) time series for each voxel in the SEQ-SIM experiment. We then implemented several pRF models in our modeling framework to computationally test our hypotheses. To test the CSS hypothesis, we used a CSS pRF model[27] as it successfully predicts subadditive responses to stimuli of different sizes in pRFs across the visual hierarchy. To test the CST summation hypothesis, we used a CST summation pRF model[51], which captures pRF responses to a large range of spatial and temporal stimulus conditions by modeling neural responses in units of visual degrees and milliseconds.

## Results

To investigate what factors affect simultaneous suppression, we designed an fMRI experiment in which participants viewed colorful patterned square stimuli in upper and lower quadrants while performing a 1-back rapid serial visual presentation (RSVP) letter task at fixation. Squares could either be presented sequentially (one after the other, in random order) or simultaneously (all at once) (Fig. 1a). For each pair of sequential and simultaneous conditions, individual square presentation is identical in size and duration within an 8-s block such that linear summation of visual inputs in space and time will generate identical responses for both sequence types. To distinguish between spatial and spatiotemporal mechanisms of simultaneous suppression, we varied square size and timing (Fig. 1b, c). Additionally, participants completed an independent retinotopy experiment[56] to delineate visual areas and estimate spatial pRF parameters in each voxel (Fig. 1d).

In each visual area, we measured BOLD responses in voxels in which pRF centers overlapped with SEQ-SIM stimulus quadrants. We then determined how spatial and temporal stimulus properties affect simultaneous suppression for each pRF across visual areas spanning ventral, lateral, and dorsal processing streams. We predict that if simultaneous suppression is of spatial origin, there will be greater suppression in higher-level than early visual areas because higher-level areas contain larger pRFs that will overlap multiple squares and also show greater spatial compression[27]. Additionally, we predict that varying square size but not timing will affect simultaneous suppression. If simultaneous suppression is of spatiotemporal origin, in addition to observing greater suppression for larger pRFs in higher-level areas, we also predict stronger suppression for long (1 s) than short (0.2 s) presentations because the former has longer sustained stimulus periods, resulting in four times fewer visual transients in the 8-s stimulus blocks than the latter (Fig. 1b).

To give a gist of the data, we first show results from example voxels in early (V1) and higher-level (VO1/2) areas of the ventral stream. These visual areas differ in overall pRF size and spatial compression: V1 pRFs are small and typically overlap only one square, whereas VO1/2 pRFs are large, typically overlap multiple squares, and have more compression than V1 pRFs.

**a** *SEQ-SIM experiment*

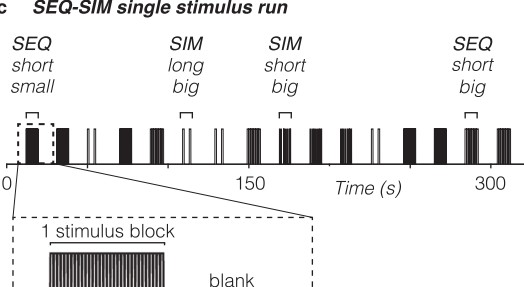

**b** *SEQ-SIM conditions*

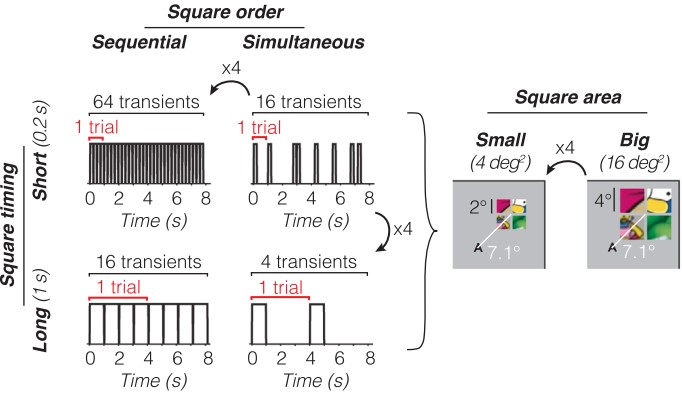

**c** *SEQ-SIM single stimulus run*

**d** *Retinotopy experiment*

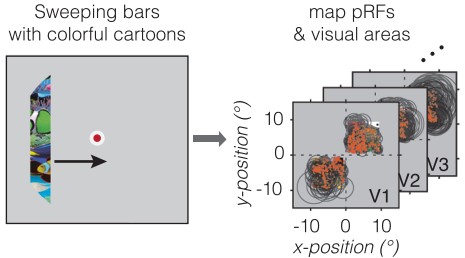

**Fig. 1 | Overview of fMRI experiments. a** SEQ-SIM experiment. Example trials for small and short stimuli. Four colorful squares were presented in the upper right and lower left quadrants, presented either sequentially in random order (left) or simultaneously (right) interspersed by blank periods to match the trial duration. Observers performed a 1-back rapid serial visual presentation (RSVP) letter task at fixation. The letter is enlarged for visibility. **b** Stimulus conditions. Square stimuli were shown either sequentially (SEQ) or simultaneously (SIM) in one of two presentation timings (0.2 s or 1 s) and in one of two sizes (4 deg$^2$ or 16 deg$^2$). The number of trials per block was adjusted to create a 4:1 ratio in the number of transients (stimulus onsets or offsets) for short vs long durations. The number of transients indicated is based on a pRF overlapping all four squares, e.g., for 1-s sequentially-presented squares there are 16 transients per block: 4 stimulus frames × 2 on/offsets × 2 trials. If a pRF overlaps only a single square (time series not shown), the number of transients will be identical for SEQ and SIM pairs. For each SEQ-SIM pairing, individual squares were shown for the same duration within a single trial (red bracket). Trials were repeated within an 8-s block (black bracket), where square content was updated for each trial. **c** Example of SEQ-SIM stimulus run. A single 332-s SEQ-SIM run contained 16 8-s pseudo-randomized stimulus blocks, interspersed with 12-s blank periods. Data are analyzed in 23-s time windows containing a pre-stimulus baseline period, one stimulus block, and a subsequent blank period (zoom). **d** Retinotopy experiment. Observers viewed bars containing cropped cartoon stimuli traversing the visual field (left, Toonotopy[56]) while fixating and performing a color change detection task at fixation[56]. The fixation dot is enlarged for visibility. Data were used to define visual areas and to select pRFs with centers overlapping stimulus quadrants in the main experiment (right). Orange dots: pRF centers. Black outlines: pRF size (two standard deviations).

## V1 voxels with small pRFs show modest to no simultaneous suppression

For a single V1 voxel with a small pRF overlapping only a single square, we find similar responses for simultaneous vs sequential presentations for the two stimulus sizes and presentation timings (Fig. 2a) as the stimulus within the pRF is identical across the two types of presentation sequences. In other words, this voxel shows no simultaneous suppression. Additionally, we observe that for this V1 voxel responses are larger for short presentations (many visual transients) vs long presentations (few visual transients) even though the total duration of stimulation across the blocks is identical. However, there is no difference in the response amplitude for small vs big squares of the same duration (left vs right panels).

To assess simultaneous suppression, we compare single voxel response amplitudes for simultaneous vs sequential presentations for each stimulus condition. No suppression will result in voxels falling on the identity line, whereas simultaneous suppression will result in voxels below the diagonal. In V1, we find that many voxels fall closely or just below the identity line (Fig. 2b, example participant and Supplementary Fig. 1, all participants) even as response levels are higher for

short vs long stimulus presentation timings. To quantify this relationship, we fit a linear mixed model (LMM) relating the simultaneous amplitude to the sequential amplitude across V1 voxels using a fixed interaction effect for conditions, and a random effect for participants (i.e., intercepts and slopes vary per participant and condition, Eq. 1). LMM slopes of 1 indicate no suppression, slopes less than 1 indicate simultaneous suppression, where smaller slopes correspond to stronger suppression levels.

Across participants, the LMM captures 86% of the variance in V1, with the following average (±SEM) suppression levels: small and long squares: $0.81 \pm 0.07$ (95% confidence interval, ($CI_{95\%}$) = 0.56–1.06), small and short squares: $0.85 \pm 0.06$ ($CI_{95\%}$ = 0.73–0.96), big and long squares: $0.85 \pm 0.09$ ($CI_{95\%}$ = 0.56–1.4), and big and short squares: $0.84 \pm 0.08$ ($CI_{95\%}$ = 0.57–1.1). Thus, V1 voxels with relatively small pRFs show modest to no simultaneous suppression.

## Strong simultaneous suppression for large pRFs in higher-level visual areas

For a single VO voxel with a large pRF overlapping all four large squares, we find lower responses for simultaneous than sequential

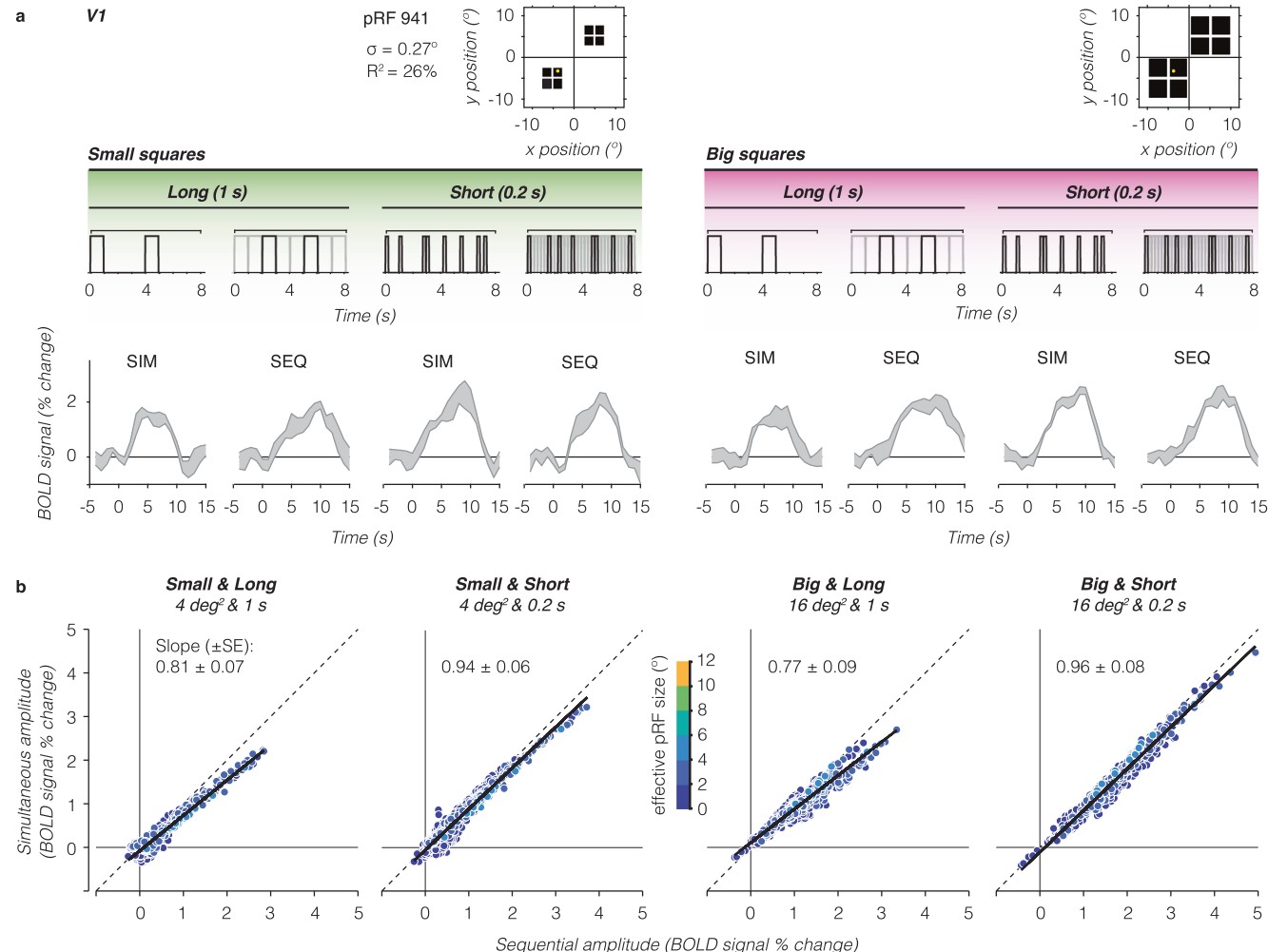

**Fig. 2 | V1 voxels show no to little simultaneous suppression. a** Example V1 voxel with small pRF overlapping a single square. The example voxel's pRF (yellow circle) is superimposed on square locations (black). Gray shaded area shows the example voxel's average BOLD time series ± standard error of the mean (SEM) across repeats for each stimulus condition. Above each time series is an example stimulus sequence for each condition in an 8-s block. Gray sequence: time series including all square stimuli. Black sequence: time series for small pRF overlapping one square. **b** Relation between BOLD amplitude (% signal change) for simultaneous vs sequential blocks, for each size/duration condition. Data include all V1 voxels from participant S3 with pRFs overlapping the square stimuli, averaged across a 9-s time window centered on the peak response. Each dot is a voxel, colored by effective pRF size from the independent retinotopy model fit ($\sigma/\sqrt{\text{CSS}_n}$). Dashed line: no suppression. Solid black line: linear mixed model (LMM) line fit for this participant's V1 data. Slope ± standard error (SE) is indicated in each panel. A slope of 1 indicates no suppression.

presentations for both square sizes and presentation timings (Fig. 3a). In other words, this voxel shows simultaneous suppression across all experimental conditions. Additionally, we observe that the overall response amplitudes of this voxel are larger for the big squares and short presentations compared to the small squares and long presentations.

We observe this pattern of results across VO voxels. Plotting the average amplitude for simultaneous vs sequential presentations, we find a linear relationship between responses to simultaneous and sequential pairings, where voxels show simultaneous suppression and the level of suppression varies across experimental conditions (Fig. 3b, example participant and Supplementary Fig. 1, all participants). This relationship is not a given, as simultaneous suppression could have tapered off with response level. Instead, our data suggests that suppression can be summarized with a single slope per visual area and experimental condition.

Quantitative analyses using a LMM ($R^2 = 97\%$) revealed significant simultaneous suppression varying with stimulus size and timing, with the following suppression levels: small and long squares: $0.40 \pm 0.07$ ($\text{CI}_{95\%} = 0.15–0.65$), small and short squares: $0.65 \pm 0.05$

($\text{CI}_{95\%} = 0.55–0.75$), big and long squares: $0.62 \pm 0.1$ ($\text{CI}_{95\%} = 0.31–0.93$), and big and short squares: $0.70 \pm 0.03$ ($\text{CI}_{95\%} = 0.54–0.87$). Notably, for stimuli of the same duration, there is stronger suppression (smaller slopes) for the small vs big squares. However, for the same square size, there is stronger suppression for long vs short presentation timings. This suggests that in VO1/2, in addition to the stimulus' spatial overlap with the pRF, timing also contributes to simultaneous suppression.

### Simultaneous suppression increases up the visual hierarchy and depends on stimulus size and presentation timing

We next quantified the relationship between responses in simultaneous vs sequential presentations across the visual hierarchy. Our data show four findings. First, in each visual area and stimulus condition, we find a linear relationship between voxels' responses to simultaneous and sequential stimuli (Fig. 4a, big and short stimuli and Supplementary Fig. 1, all conditions). Second, when quantifying this linear relationship by its slope, we find that simultaneous suppression is prevalent at the voxel level in almost every visual area across participants. Third, across all stimulus conditions, we find that suppression levels progressively increase from early visual areas (V1 to V2 to V3) to

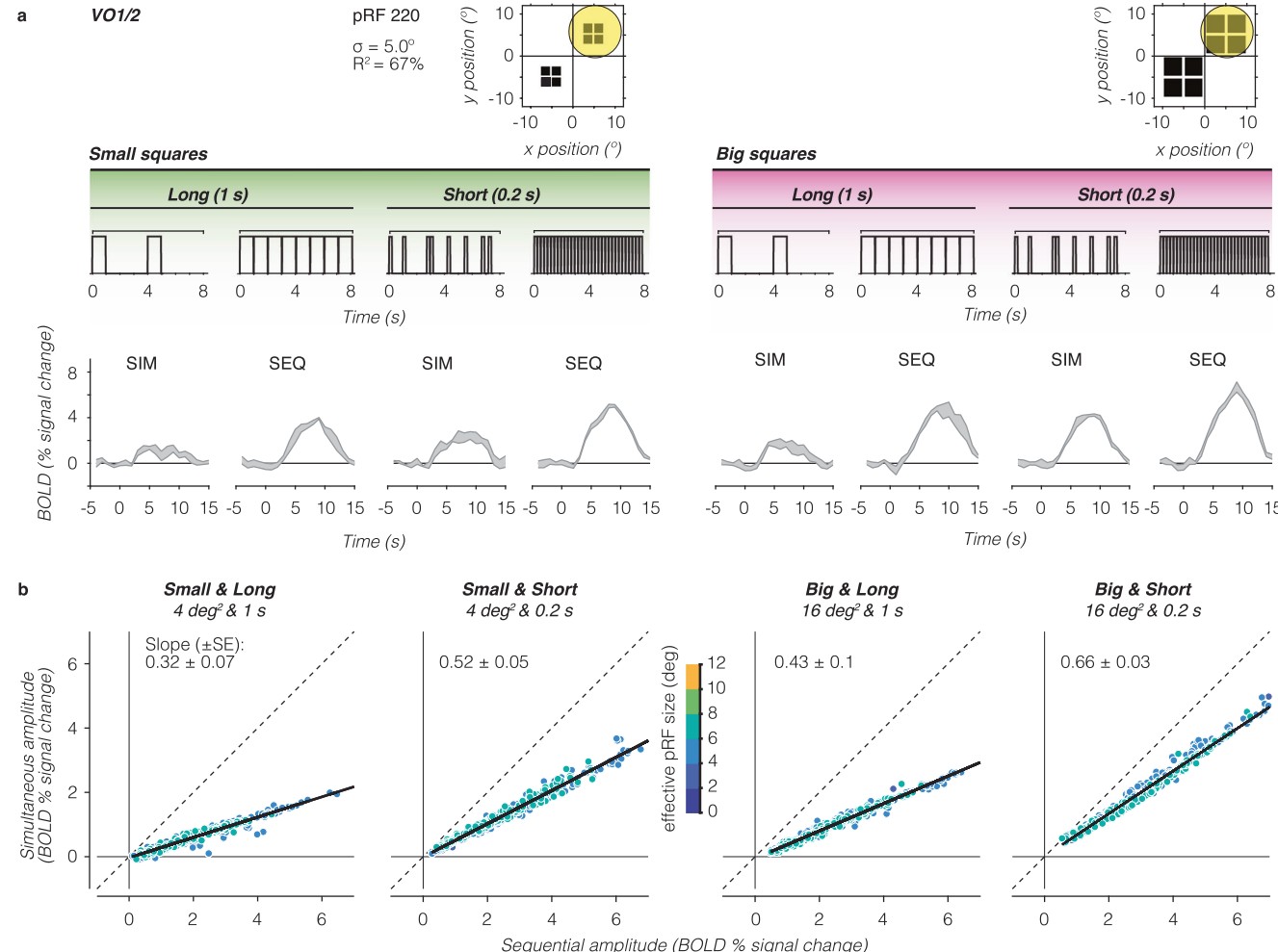

**Fig. 3 | Individual VO1/2 voxels with large pRFs show strong simultaneous suppression effects.** Same layout as Fig. 2, but for higher-level visual area VO1/2. Data are from participant S3. **a** Example time series of a VO1 voxel. The example voxel has a large pRF that covers all four squares of both sizes (yellow circle). **b** Simultaneous vs sequential BOLD amplitude for all voxels in VO1/2. Dashed line: no suppression. Solid black line: LMM line fit for this participant's VO1/2 data. Slope (± SE) is indicated in each panel.

intermediate areas (hV4, LO1/2, and V3A/B), with the strongest simultaneous suppression in TO1/2 (Fig. 4b and Supplementary Table 1). Fourth, up the visual hierarchy, simultaneous suppression levels depend on stimulus conditions. In particular, higher-level visual areas show stronger suppression for long vs short presentation timings, and stronger suppression for small vs big square sizes. A two-way repeated measures ANOVA revealed significant effects of visual area ($F_{(8)} = 23$, $p = 7.3 \times 10^{-27}$) and stimulus condition ($F_{(3)} = 27$, $p = 2.3 \times 10^{-15}$) on suppression slopes. There was no significant interaction between stimulus condition and visual area (Supplementary Table 2; post-hoc Bonferroni-corrected $t$-tests).

The increasing suppression levels across the visual hierarchy are in line with our prediction that simultaneous suppression will be stronger in visual areas that have larger pRF sizes. This relationship is evident at the level of entire visual areas (Fig. 4b), but not across voxels within an area (Fig. 4a). Within an area, we find similar suppression levels for voxels with pRFs that drastically vary in size (e.g., VO1/2), yet their level of suppression is predicted by a single line. Thus, while pRF size is an important predictor of simultaneous suppression at the level of an entire visual area, our data suggest that by itself, summation within pRFs that vary in size is insufficient to explain different suppression levels observed across stimulus conditions. Together, these results reveal robust simultaneous suppression at the individual voxel

level that depends both on pRF size alongside stimulus size and timing parameters.

To understand how much of the observed suppression in higher-level visual areas is accumulated from earlier visual areas, we compare suppression slopes between pairs of consecutive visual areas within a processing stream. One possibility is that suppression monotonically accumulates up the visual hierarchy irrespective of stimulus condition (e.g., a consistent difference in slopes between consecutive areas). Alternatively, suppression may increase until a certain processing stage and then plateau (e.g., when the average pRF size within a visual area is large enough to encompass all square stimuli). Contrary to these predictions, we find that the difference in suppression levels between consecutive visual areas varies by stimulus condition, and suppression levels do not increase consistently across the visual hierarchy nor plateau (Supplementary Fig. 2a). Additionally, observed differences in suppression levels between consecutive visual areas do not show a clear relationship with differences in pRF size (Supplementary Fig. 2b) or differences in spatiotemporal compression within pRFs (Supplementary Fig. 2c). These results suggest that there is some accumulation of simultaneous suppression up the visual hierarchy, but that accumulation alone cannot fully explain the observed simultaneous suppression levels in higher visual areas.

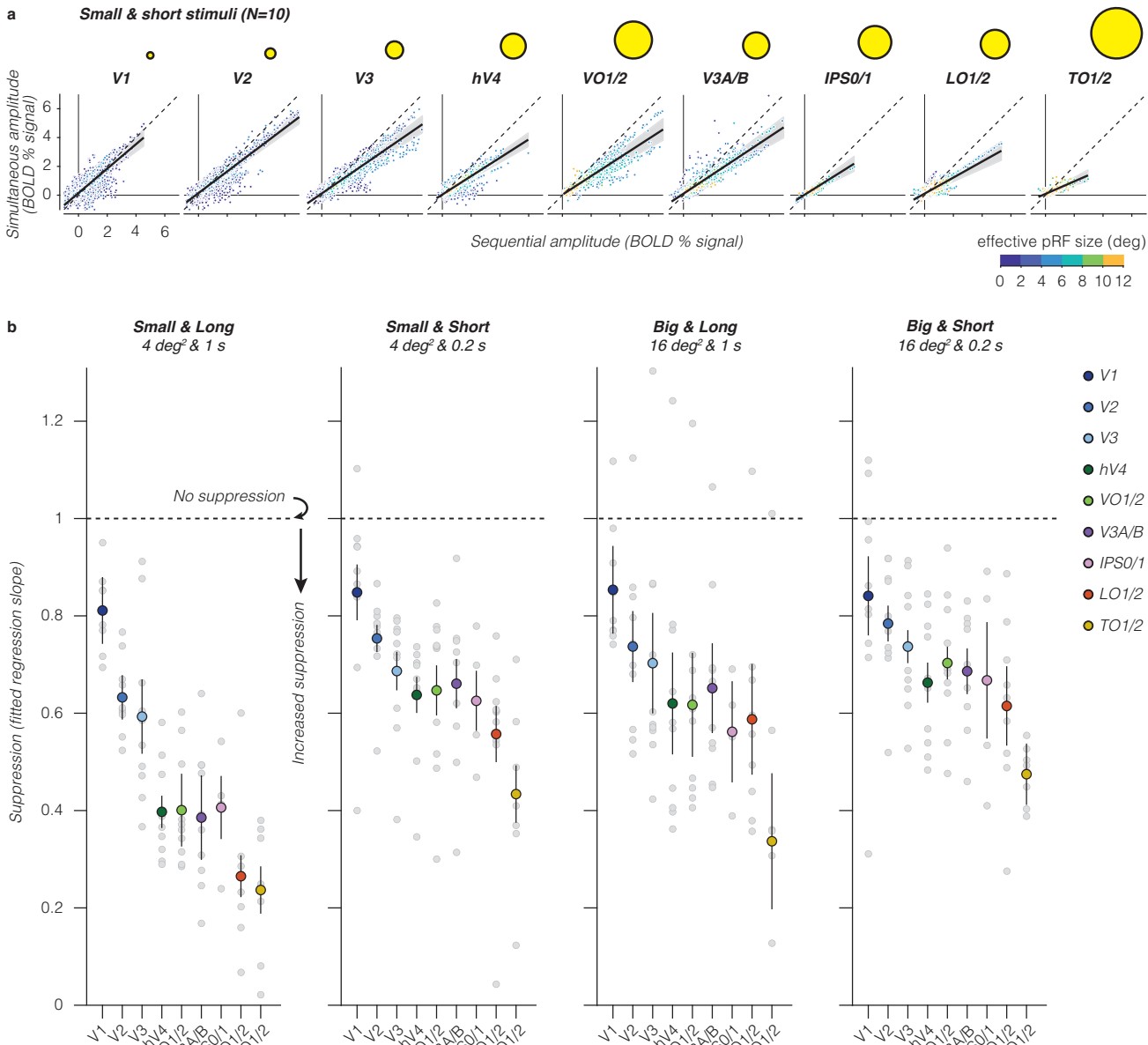

**Fig. 4 | Simultaneous suppression increases up the visual hierarchy. a** Sequential vs simultaneous BOLD amplitude of individual voxels for the small and short stimulus condition. Each point is a voxel, colored by effective pRF size estimated from independent retinotopy data. Each panel shows data of all ten participants. Black solid line: LMM fit (average across participants). Dashed line: identity line. Shaded area: CI$_{95\%}$ across ten participants. Yellow circles: illustration of average effective pRF size per visual area, ranging from 1° in V1 to 7.8° in TO1/2. **b** Suppression levels for each stimulus condition and visual area. Slopes are derived from LMM fit to simultaneous vs sequential average BOLD amplitude data from all ten participants, for each visual area. A slope of 1 indicates no suppression. Smaller slopes indicate increased suppression. Large colored dots: group average of a visual area. Error bars: SEM across participants. Light gray dots: individual participant slopes (random effects). Early visual areas are in blue colors (V1: indigo, V2: dark blue, and V3: light blue), ventral visual areas in green colors (hV4: dark green and VO1/2: light green), dorsal visual areas are in purple colors (V3A/B: purple and IPS0/1: pink), and lateral visual areas are in warm colors (LO1/2: red and TO1/2: yellow).

## A spatiotemporal pRF modeling framework to predict simultaneous suppression at the single voxel level

To gain insight into the stimulus-driven computations that give rise to different levels of simultaneous suppression at the voxel level, we developed a computational framework that predicts the neural population response in each voxel from its pRF given the frame-by-frame stimulus sequence of the SEQ-SIM experiment (Fig. 5). To capture the brief nature of the stimuli and the neural response, both stimulus sequence and predicted pRF responses have millisecond resolution. This neural pRF response is then convolved with the hemodynamic response function (HRF) to predict the voxel's BOLD response and downsampled to 1-s resolution to match the fMRI

acquisition resolution (Fig. 5a). Crucially, for each voxel, we use a single pRF model and the stimulus sequence of the entire SEQ-SIM experiment to predict its time series across all stimulus conditions at once. For all tested pRF models, the spatial parameters of each voxel's pRF are identical and estimated from the independent retinotopy experiment (Fig. 1d).

We test three pRF models. First, a CST summation pRF model[51] (Fig. 5b) to quantitatively examine if compressive spatiotemporal summation within pRFs can predict simultaneous suppression across all stimulus manipulations. The CST pRF model contains three spatiotemporal channels that have the same spatial pRF (2D Gaussian) but different neural temporal impulse response functions

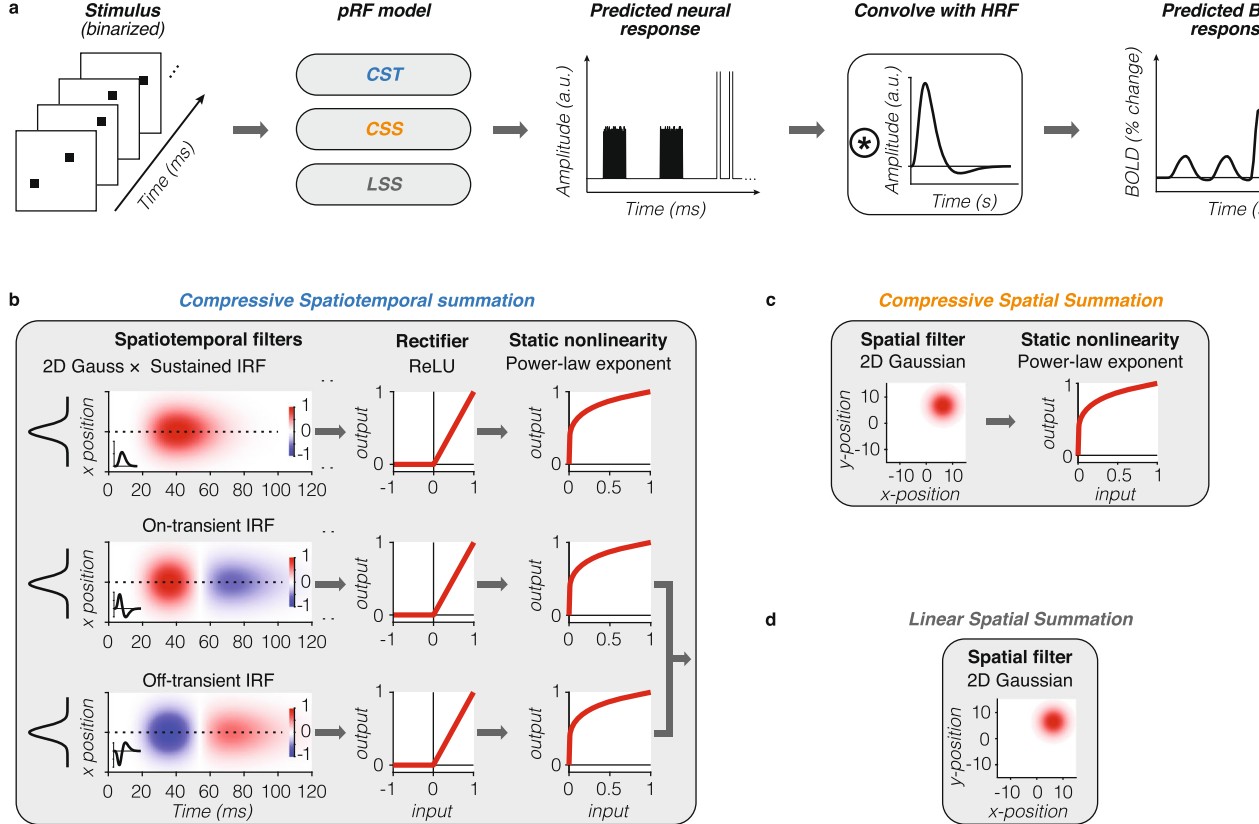

**Fig. 5 | Computational modeling framework. a** Model overview. From left to right: given a binarized stimulus sequence and pRF model, the neural response is predicted at millisecond time resolution. This neural response is convolved with HRF to predict the BOLD response. After the convolution with the HRF, data are downsampled to 1-s temporal resolution (TR in SEQ-SIM experiment). a.u.: arbitrary units. **b**–**d** Tested pRF models. For each voxel, spatial pRF parameters are identical for all models and estimated from the independent retinotopy experiment (Fig. 1d). Both CSS and LSS models sum linearly over time. For simulated pRF model predictions, see Supplementary Fig. 3. **b** Compressive spatiotemporal (CST) summation pRF model[51]. Temporal pRF parameters are default parameters from ref. 41.

Static power-law exponent parameter (< 1) is the same for all three spatiotemporal channels and fitted to each voxel's SEQ-SIM data. The overall predicted BOLD response by the CST model is the weighted sum of the sustained and combined transient channels. Red-blue color bar indicates the normalized amplitude of the impulse response function (IRF). ReLU: rectified linear unit. **c** Compressive spatial summation (CSS) pRF model[27]. 2D Gaussian followed by a static compressive nonlinearity (power-law exponent < 1, estimated from retinotopy data). **d** Linear spatial summation (LSS) pRF model[18]. LSS pRFs sum linearly across space and time by computing the dot product between the binarized stimulus frame and the 2D Gaussian pRF.

(IRFs): a sustained, on-transient, and off-transient channel that captures stimulus duration, onsets, and offsets; neural IRFs use default temporal pRF parameters from ref. 41. These spatiotemporal filter outputs are rectified and subjected to a compressive static nonlinearity, which produces subadditive spatiotemporal summation for both sustained and transient channels.

Second, we implement a CSS pRF model[27] (Fig. 5c) to quantitatively test if subadditive spatial summation alone can explain simultaneous suppression. The CSS model is a 2D Gaussian followed by a compressive static nonlinearity, and is successful in predicting spatial subadditivity in voxels with larger pRFs beyond V1.

Third, we implement a linear spatial summation pRF model (LSS[18]) (Fig. 5d) to quantitatively test if small voxels that show little to no simultaneous suppression, such as those in V1, can be predicted by linear summation in space and time. The LSS pRF is a 2D Gaussian and sums over the stimulus linearly in time and across space. This model was also used to validate our experimental design as it predicts that, irrespective of pRF size, linear summation of stimuli in paired SEQ-SIM conditions should result in the same response, i.e., no simultaneous suppression.

We test these pRF models for four main reasons. First, they describe a neural mechanism with a receptive field restricted to part of the visual field; this restriction is needed to test the impact of stimulus location and size. Second, the identical spatial pRF across models and

the similar static nonlinearity implementation for compressive models (CST and CSS) allow for informative comparisons between models. Third, both compressive models have the potential to predict simultaneous suppression within this stimulus regime. Fourth, CST and CSS models have been successful in providing a comprehensive explanation for subadditive visually-driven responses across visual cortex[27,51].

## Comparing pRF model performance in predicting observed SEQ-SIM data

For each voxel, we generate three predicted BOLD responses, one for each tested pRF model (CST, CSS, and LSS; see Supplementary Fig. 3 for example pRF model predictions). We fit each model using split-half cross-validation and quantified the cross-validated variance explained (cv-$R^2$) for each voxel. This provides a principled and unbiased way to test the hypotheses.

For our example small V1 pRF, both spatial models (LSS and CSS) predict the same BOLD response for sequential and simultaneous pairs (Fig. 6a, bottom and middle rows). This is because the pRF covers only one small square, and consequently, the spatial summation is identical across SIM and SEQ presentations. Comparing predictions to data, both LSS and CSS models capture the voxel's response to long stimulus conditions, but underpredict the voxel's response for short stimulus conditions, resulting in the same cv-$R^2$ of 44% for this V1 voxel. In comparison, the CST pRF model best captures the response pattern

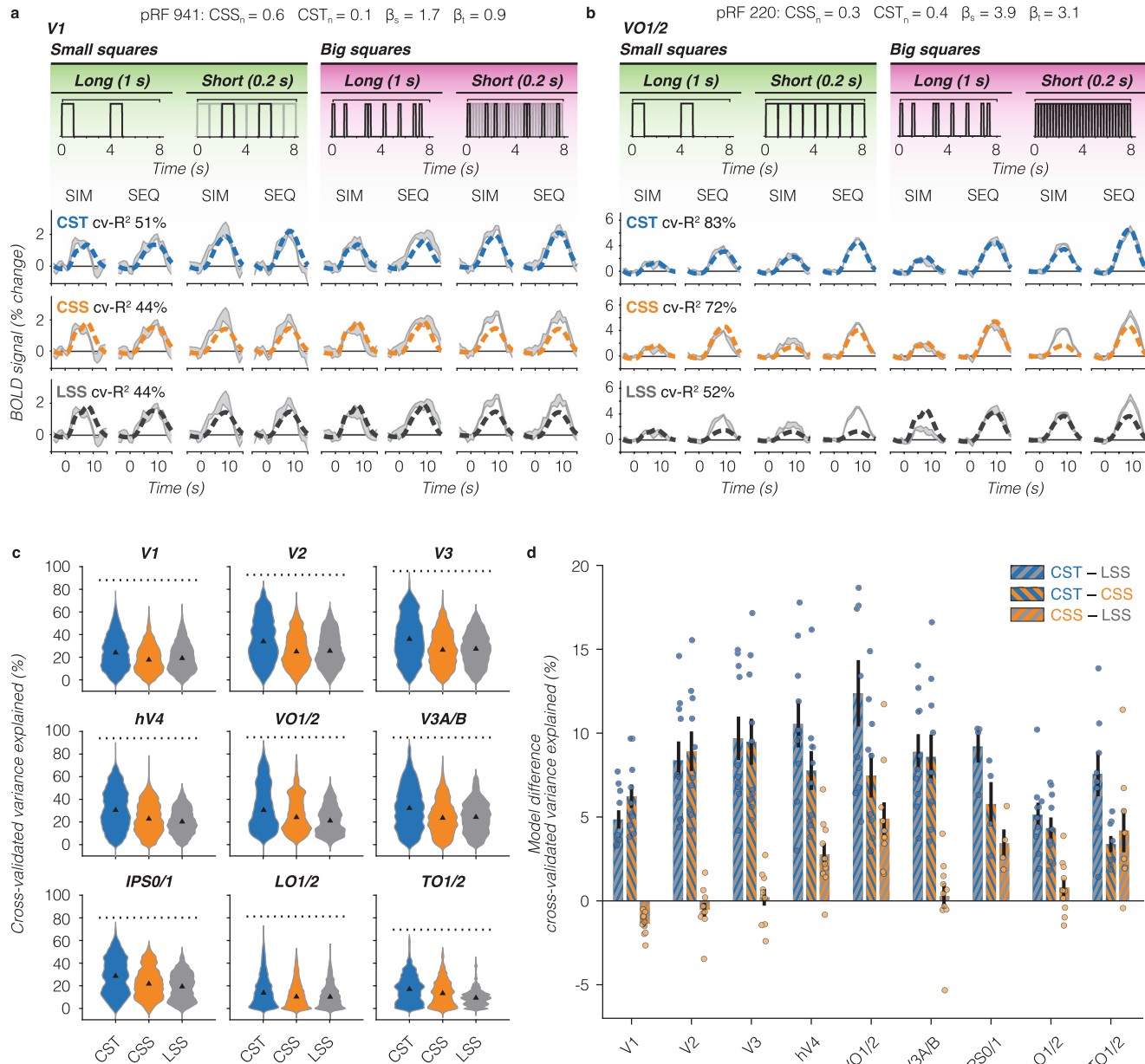

**Fig. 6 | Comparison of performance across pRF models. a** V1 example voxel time series. Gray shaded area: average ± SEM across repeats for each stimulus condition. Data are from the same voxel as in Fig. 2a repeated for each row. PRF model fits are shown in dashed lines. Split-half cross-validated variance explained (cv-$R^2$) is computed by fitting the predicted time series to the average of odd runs and applying the model fit to the average of even runs and vice versa. Blue: CST summation model (top row). Orange: CSS model (middle row). Black: LSS model (bottom row). **b** VO1/2 example voxel. Data are from the same voxel as in Fig. 3a repeated for each row. Same color scheme as panel A. **c** Distribution of voxel-level

cv-$R^2$ for each pRF model, all ten participants. Triangle: median. Dotted line: noise ceiling computed from max split-half reliability across participants. Blue: CST. Orange: CSS. Gray: LSS. Since a number of voxels varies per participant and visual area, we assure equal contribution of each participant by resampling data 1000 times of each participant's visual area. **d** Pairwise model comparison for each visual area. Bars: average across ten participants of the voxelwise difference in cv-$R^2$ between two pRF models. Error bars: SEM across ten participants. Individual dots: average difference for each participant. Blue-gray: CST vs LSS. Blue-orange: CST vs CSS. Orange-gray: CSS vs LSS.

across all stimulus conditions (cv-$R^2$ = 51%), predicting no suppression and larger BOLD amplitudes for short than long stimulus conditions (Fig. 6a, top row).

When pRFs are large and cover multiple stimuli, like the example VO1/2 voxel, the LSS pRF model predicts larger responses for big than small squares, slightly higher responses for long than short presentations, and identical responses for sequential and simultaneous pairs. As such, it fails to predict the observed simultaneous suppression in all conditions (Fig. 6b, bottom row). On the other hand, the CSS pRF model predicts simultaneous suppression because of spatial

subadditivity, as well as a modest increase in response with stimulus size (Fig. 6b, middle row). Like the LSS model, the CSS model predicts slightly larger responses for the long than short presentations of a given sequence type (SIM/SEQ). Consequently, the CSS model predicts simultaneous suppression well for long presentations across stimulus sizes, but overpredicts simultaneous suppression for short presentations. In contrast, the CST pRF model best predicts all stimulus conditions for this example voxel: it shows simultaneous suppression, slightly larger responses for big vs small stimulus sizes, and larger responses for short vs long presentation timings (Fig. 6b, top row).

Across all voxels and visual areas, we find that the CST pRF model best predicts the observed data in the SEQ-SIM experiment (Fig. 6c, d). The CST pRF model explains more cv-$R^2$ than LSS and CSS pRF models and approaches the noise ceiling in V3 and higher-level visual areas (Fig. 6c, dotted line). A two-way repeated measures ANOVA revealed significant effects of pRF model ($F(2) = 2.6 \times 10^3$, $p = 10^{-209}$) and region of interest (ROI) ($F(8) = 3.4 \times 10^3$, $p = 10^{-209}$) on cv-$R^2$, as well as a significant interaction between pRF model and ROI ($F(2,8) = 65$, $p = 2.8 \times 10^{-209}$) (post-hoc Bonferroni-corrected $t$-tests are reported in Supplementary Table 3). On average, the increase in cv-$R^2$ for the CST model compared to the other models ranges from ~5% in V1 to ~12% in VO1/2 (Fig. 6d). Beyond early visual cortex, the CSS model outperforms LSS, but in V1 the LSS model slightly ($M = 1.4\%$, $SE = 0.24\%$) and significantly ($p = 2.7 \times 10^{-8}$, $CI_{95\%} = 0.80-2.0\%$) explains more cv-$R^2$ than the CSS model. These results suggest that V1 voxels largely sum linearly in space, but nonlinearly in time. However, across the visual hierarchy, CST summation provides a more comprehensive explanation of the empirical data.

## To what extent do pRF models predict simultaneous suppression across visual cortex and stimulus conditions?

To understand the underlying neural computations that generate simultaneous suppression, we use pRF models to predict the level of simultaneous suppression in each voxel and condition of the SEQ-SIM experiment. Then, we compare the model-based simultaneous suppression against the observed suppression (Fig. 7, shaded gray bars).

The CST model best captures simultaneous suppression across visual areas and stimulus conditions as its predictions are largely within the range of data variability (Fig. 7, compare blue circles to shaded gray bars). Specifically, the CST model predicts (i) progressively increasing simultaneous suppression across the visual hierarchy, (ii) stronger suppression for longer than shorter presentation timings for squares of the same size, and (iii) weaker suppression for bigger

than smaller squares of the same timing. The CST model performs similarly with pRF parameters that are optimized using data from the independent spatiotemporal retinotopy experiment[51] (Supplementary Figs. 8 and 9).

The CSS model captures the progressively stronger simultaneous suppression across the visual hierarchy and the observed simultaneous suppression for the long stimuli in a few visual areas (V3A/B, IPS0/1, and TO1/2), but fails to predict suppression for short stimuli and generally overpredicts the level of suppression (Fig. 7, orange circles). In other words, the CSS model predicts much stronger simultaneous suppression levels than observed, as model predictions are consistently below the data. This overprediction is largest for short presentation timings in early (V1–V3) and ventral visual areas (hV4 and VO1). One reason for this mismodeling error is that the CSS model does not encode visual transients: it predicts stronger simultaneous suppression for small than big sizes but predicts similar simultaneous suppression for long and short presentations of the same square size.

Finally, and as expected, the LSS model does not predict simultaneous suppression altogether. This is because the LSS model sums visual inputs linearly in space and time, and we designed our experiment such that each square is shown for the same duration and location in sequential and simultaneous conditions. Therefore, the LSS model predicts the same responses for sequential and simultaneous stimulus pairings and consequently no suppression (Fig. 7, black open circles). For the big and long squares, the LSS model predicts slightly higher responses for simultaneous vs sequential presentations. We attribute this to our experimental design, which has different inter-stimulus intervals (ISIs) of individual squares between sequential and simultaneous blocks, see "Methods−LSS pRF model").

Thus far, we primarily focused on two compressive pRF models (CST and CSS) and found that CST pRFs best predict simultaneous suppression. However, this does not rule out the possibility that

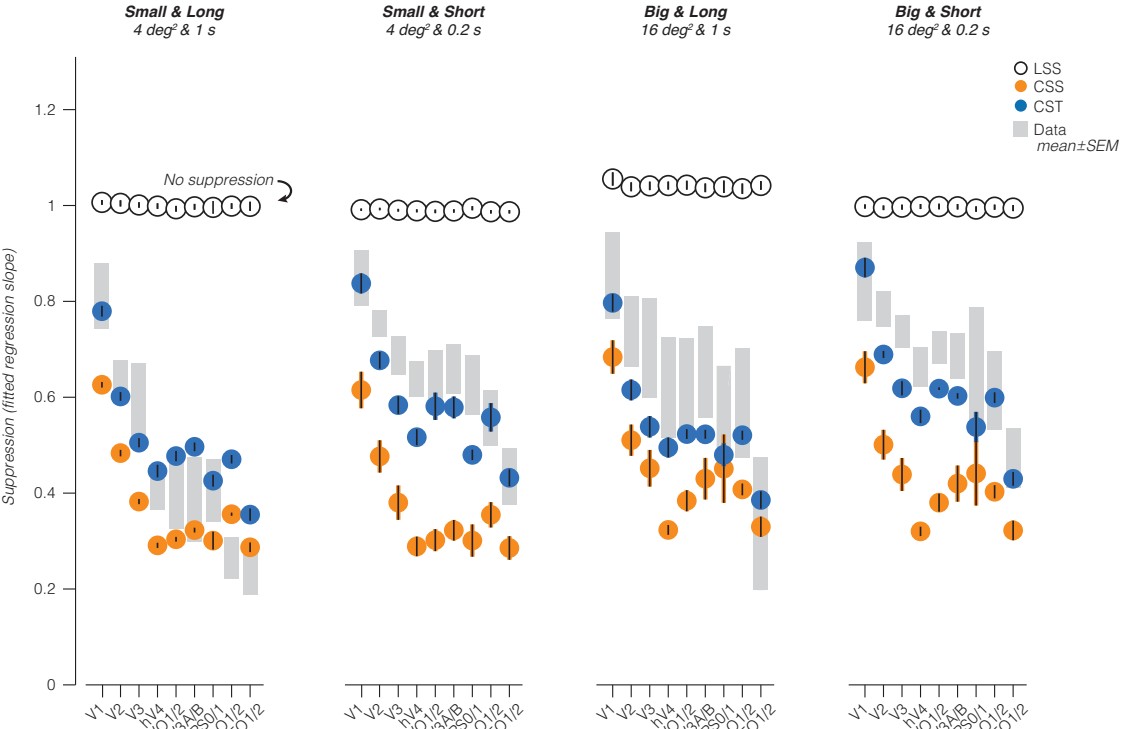

**Fig. 7 | Model-based prediction of simultaneous suppression vs observed simultaneous suppression.** Shaded gray bars: observed suppression levels in data, average ± SEM across ten participants (same as Fig. 4b). Black open circles: LSS pRF model. Orange filled circles: CSS pRF model. Blue filled circles: CST summation pRF model. Model-based points and error bars show average and SEM across all ten participants.

other types of subadditive mechanisms within pRFs can account for simultaneous suppression. One such mechanism is center-surround suppression, which is most prevalent in the early stages of visual processing (retina[57], LGN[24], and V1[58]). To test this possibility, we simulated difference of Gaussians (DoG) pRFs[48] for voxels in V1 through hV4. In these visual areas, DoG pRFs have a positive center that encompasses one or few squares while the larger, negative surround covers more squares, which may suppress the overall response in simultaneous conditions (Supplementary Fig. 7a). Yet, we find that DoG pRFs predict no simultaneous suppression in V1 through hV4 for our stimuli (Supplementary Fig. 7b). Instead, like the LSS model, DoG pRFs sum linearly over the spatial and temporal extent of the stimulus, resulting in similar responses across SIM and SEQ conditions.

A second potential subadditive mechanism is delayed normalization[42,44]. To test this mechanism, we implement a delayed normalization spatiotemporal (DN-ST) pRF model for seven participants who completed a separate spatiotemporal retinotopy experiment[51]. Like the CST model, the DN-ST model captures the increase in simultaneous suppression across the visual hierarchy, as well as differences in suppression levels with stimulus size and timing (Supplementary Fig. 8a). A two-way repeated measures ANOVA reveals significant effects of pRF model ($F(2) = 1.1 \times 10^2$, $p = 6.3 \times 10^{-47}$) and ROI ($F(7) = 1.3 \times 10^3$, $p = 10^{-47}$) on cv-$R^2$ (Supplementary Fig. 8b), where the DN-ST model with optimized pRF parameters predicts overall less cv-$R^2$ than either CST model: 3.9% less cv-$R^2$ than CST pRFs with fixed parameters and 1.5% less cv-$R^2$ than CST pRFs with optimized parameters. In particular, DN-ST pRFs tend to underpredict the level of simultaneous suppression for short stimulus timings (0.2 s) (Supplementary Fig. 9a). The ANOVA also indicates a significant interaction between the pRF model and ROI ($F(2,7) = 5.0$, $p = 1.6 \times 10^{-9}$), where both CST models perform significantly better than the DN-ST model in almost all visual areas, and the main CST pRF model explains slightly but significantly more cv-$R^2$ than the optimized CST pRF model in visual areas V1, hV4, V3A/B, LO1/2, and TO1/2 (Supplementary Table 4, post-hoc Bonferroni-corrected $t$-tests).

Together, these model comparisons suggest that accounting for spatiotemporal nonlinearities rather than just spatial nonlinearities is necessary for predicting simultaneous suppression across a variety of spatial and temporal stimulus conditions.

## What intrinsic pRF components drive the observed simultaneous suppression?

To elucidate what pRF components predict the varying levels of simultaneous suppression across the visual system, we examine the relationship between the average suppression level and CST pRF model parameters. We find that simultaneous suppression increases with pRF size, spatiotemporal compression ($CST_n$), and necessitates contributions from both sustained and transient temporal channels (Fig. 8). Visual areas with larger pRF sizes tend to show stronger simultaneous suppression levels (smaller slopes, Pearson's correlation $r = -0.72$, $CI_{95\%} = -0.81$ to $0.59$, $p = 10^{-5}$) (Fig. 8a). Likewise, visual areas with stronger spatiotemporal compression within pRFs (smaller exponents) are linked to stronger simultaneous suppression levels (Pearson's $r = 0.65$, $CI_{95\%} = 0.50-0.76$, $p = 10^{-5}$) (Fig. 8b).

Lastly, we find that across visual areas, both sustained and transient channels contribute to predicting single voxel BOLD responses, as their β-weights are similar (no significant difference in β-weights across channels) (Fig. 8c). These results indicate that both sustained and transient channels are needed to predict simultaneous suppression across different stimulus size and timing conditions.

Examining the relationship between simultaneous suppression and optimized CST model parameters underscores our findings that larger pRF sizes, stronger compressive nonlinearity (i.e., smaller exponents), and contributions of both sustained and transient channels are important for predicting the level of simultaneous suppression across the visual hierarchy, whereas time constant parameters do not systematically co-vary with observed suppression levels (Supplementary Fig. 9b–e). The DN-ST model shows similar a relationship as the CST model, where increased levels of simultaneous suppression are predicted by larger pRF sizes and stronger spatiotemporal compression within the pRF via larger semi-saturation constants in the denominator, smaller exponents, and increased exponential decay time constants (Supplementary Fig. 9f–i). These results suggest that simultaneous suppression can be predicted by more than one implementation of a CST mechanism (static nonlinearity vs delayed divisive normalization).

Because the static nonlinearity in CST pRFs is applied to the output of spatiotemporal channels, the compressive nonlinearity is of a spatiotemporal nature and cannot be separated into spatial and temporal dimensions. Nevertheless, we can gain insight into the

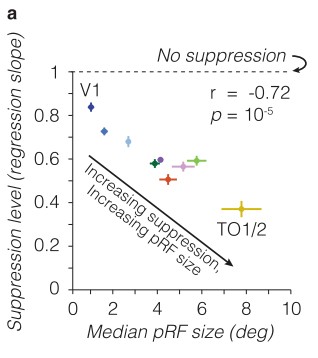
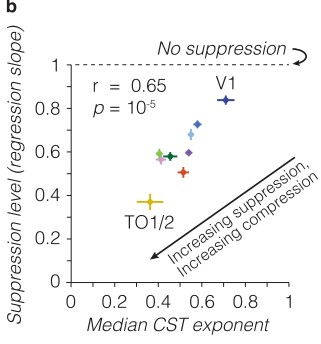
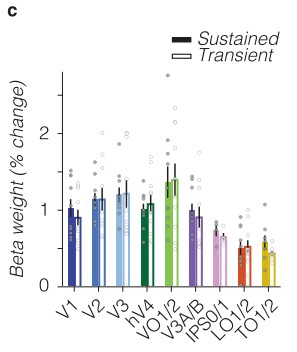
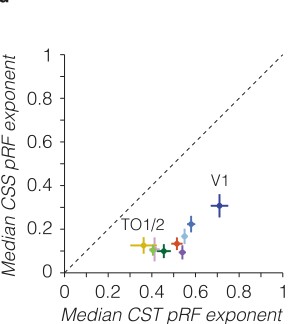

● V1  ● V2  ● V3  ● hV4  ● VO1/2  ● V3A/B  ● IPS0/1  ● LO1/2  ● TO1/2

**Fig. 8 | Simultaneous suppression depends on pRF size, compressive exponent, and contributions from both sustained and transient channels.** In all panels: dots/bars show the average across ten participants. Error bars: SEM across ten participants. **a** Simultaneous suppression level vs median pRF size. **b** Simultaneous suppression level vs median CST pRF exponent. For effective size and exponent pRF parameters, we first computed the median across pRFs of a visual area for each participant, as compressive exponent values in V1 and V2 voxels are not normally distributed (see Supplementary Fig. 4), then we calculated the average median value across participants. Pearson's correlation ($r$) is computed using individual participant data. $p$: $p$-value. **c** Average β-weights of sustained and transient channels in CST pRF model. Beta weights are averaged first within a participant's visual area, then averaged across participants per visual area. Colored bars: sustained channel. White bars with a colored outline: combined transient channel. Dark gray dots: individual participant data for the sustained channel. Light gray dots: individual participant data for the combined transient channel. Differences between sustained and combined transient channels are not significant. **d** Median exponent pRF parameters for CSS vs CST model. A dashed line indicates an equality line.

different contributions of spatial versus spatiotemporal compression by comparing the exponent across the main CST and CSS pRF models. We find that across all visual areas, CSS pRFs have smaller exponents (resulting in more compression) than CST pRFs with a fixed time constant (Fig. 8d). This overly strong compression by the CSS model likely explains its mismodeling of the short stimuli, where it predicts too much suppression (Fig. 7). Interestingly, while CST pRFs with optimized spatiotemporal parameters have similarly small exponents like CSS pRFs (Supplementary Fig. 9c), the CST pRFs are more accurate than CSS pRFs in predicting simultaneous suppression across stimulus conditions (Supplementary Fig. 9a).

Overall, these results suggest that both spatial and temporal nonlinearities within pRFs are necessary to account for the observed simultaneous suppression, and ultimately interact.

## Discussion

Simultaneous suppression is a decades-old, yet perplexing neurophysiological phenomenon: Why is the response to multiple stimuli presented simultaneously substantially lower compared to the response to the same stimuli presented sequentially? Here, we combined a new experimental design, varying stimulus size and presentation timing, with an innovative spatiotemporal pRF modeling framework to elucidate the stimulus-driven computations that may give rise to simultaneous suppression in individual voxels. Our results show that the level of simultaneous suppression depends not only on the spatial overlap between stimuli and the pRF, but also on the timing of stimuli and the number of visual transients. Furthermore, we find that compressive (subadditive) spatiotemporal computations by pRFs predict simultaneous suppression in each voxel across the visual hierarchy, and across various experimental conditions. These findings suggest that stimulus-driven CST computations by pRFs generates simultaneous suppression and necessitates a rethinking of the neural mechanisms involved in simultaneous suppression.

By investigating simultaneous suppression under a computational lens, measuring and predicting each voxel's pRF response independently, we provide a mechanistic explanation on how the spatial overlap between the stimulus and pRF drives simultaneous suppression at the single voxel level. This confirms the longstanding hypothesis that the overlap between the receptive field and stimuli matters[8,9,12]. Additionally, we show that increasing simultaneous suppression up the visual hierarchy is predicted by both the progressive increase in pRF size and strength of spatiotemporal compression.

Crucially, we are able to explain a wide range of simultaneous suppression levels by stimulus-driven computations within pRFs alone, which necessitates a rethinking of the neural processing underlying simultaneous suppression. Thus, we propose a new idea that simultaneous suppression is a consequence of simple, stimulus-driven spatiotemporal computations rather than a result of stimuli competing for limited neural resources within receptive fields and prioritized by task demands. As our computational framework uses a stimulus-referred encoding model, it has predictive power. This allows future research to make new predictions about suppression levels for any stimulus sequence. The framework is also modular and can be expanded to computationally operationalize the effects of stimulus content, context, and task demands on simultaneous suppression.

Consistent with previous work[7–9,12], our data show that simultaneous suppression increases up the visual hierarchy and is particularly strong in ventral visual areas (hV4 and VO1/2). Notably, we find that not only stimulus size and location, but also stimulus timing and number of visual transients affect the level of simultaneous suppression: for stimuli of the same size, longer timings (1 s) with fewer transients generated stronger suppression levels than shorter timings (0.2 s) with more transients.

Contrary to prior studies[8–12], we find moderate levels of suppression already in V1, despite its small receptive fields. This may be because of multiple differences between studies. First, differences in the level of analysis: we quantify simultaneous suppression at the voxel level vs ROI level in prior fMRI studies[8–12]. Second, differences in which pRFs are analyzed: we include all pRFs that overlap the stimuli, including small pRFs that partially overlap multiple squares, but prior electrophysiology studies used stimuli that completely overlap with neurons' receptive fields[5–7]. Third, differences in stimulus timing: prior studies used a single stimulus timing (0.25 s per stimulus)[8–12], which is similar to our short stimuli, for which we find weaker levels of simultaneous suppression.

To test whether CSS or CST summation can predict simultaneous suppression across experimental conditions, we compared multiple pRF models. Overall, the CST pRF model provides a comprehensive explanation for simultaneous suppression across voxels spanning the ventral, dorsal, and lateral visual processing streams, as well as stimuli varying in size and presentation timing. This high performance of the CST model across all visual areas is not a given, as different models could have better predicted certain visual areas or processing streams.

Spatial pRF models captured some, but not all aspects of the observed simultaneous suppression. For instance, CSS pRFs are able to predict simultaneous suppression for long (1 s) but not shorter (0.2 s) presentation timings in several visual areas. As LSS, DoG, and CSS pRF models are developed for stimulus durations and intervals that evoke BOLD responses that approximately sum linearly in time, these models are limited because they do not account for visual transients. This assumption of temporal linearity is not only a limitation of the spatial pRF models we tested, but of any other pRF model that sums linearly over the stimulus duration, such as linear spatiotemporal pRF models[49] or other center-surround pRFs[50].

Likewise, we show that other mathematical forms of subadditive spatiotemporal summation, a DN-ST pRF model, can predict simultaneous suppression across stimulus conditions, as well as across visual areas. When inspecting pRF parameters of the tested spatiotemporal pRF models (CST and DN-ST), we find that pRF size and compressive nonlinearities that incorporate visual transients at the neural level are crucial for predicting simultaneous suppression.

While our spatiotemporal pRF models outperform spatial pRF models in predicting simultaneous suppression across stimuli size and timing, CST and DN-ST pRF models did not capture all spatiotemporal nonlinearities. For instance, for long stimuli, both CST and DN-ST models tend to overpredict suppression in early visual areas. Future research may improve CST and DN-ST model performance by optimizing parameters of both neural and hemodynamic temporal IRFs in each voxel[51]. Additionally, the estimation of some pRF model parameters is more sensitive to the experimental design and HRF variability (e.g., pRF time constant) than others (e.g., pRF position)[51]. Therefore, we stress that it is important to consider how the experimental design may affect pRF parameter estimates and subsequent model performance.

We are not the first to consider temporal aspects of BOLD responses in models of the human visual system. Prior studies have suggested other hemodynamic[45–47] and neural[35,39–42,48–50] IRFs to capture BOLD temporal nonlinearities (see review[59]). Notwithstanding the success of these models, only the recent development of a CST pRF model[51] with neural IRFs in units of visual degrees and milliseconds provided us with the opportunity to examine what subadditive spatiotemporal computations contribute to simultaneous suppression for the following reasons. First, a successful model needs to account for neural nonlinearities. We believe that the observed nonlinearities are of neural rather than hemodynamic origin, as electrocorticography and single-unit recordings show that neural responses to brief visual stimuli evoke strong visual transients and are nonlinear[42]. In a recent study, we have shown that implementing such neural nonlinearities in a computational model rather than optimizing hemodynamic responses is necessary to predict BOLD temporal nonlinearities to

brief stimuli as in the present study[51]. Second, to capture visual transients in rapid succession, the model requires neural IRFs with millisecond precision and a 50–200 ms response window rather than a 1–4 s window as afforded by hemodynamic models[47,49]. Third, the model also requires a spatial pRF. While prior studies have modeled neural IRFs with millisecond time resolution[35,39–42], without a spatial component these models are unable to predict differences in responses to one vs multiple stimuli covering a pRF.

A key insight from our study is that both increasing pRF size and stronger spatiotemporal compression contribute to increasing levels of simultaneous suppression up the visual processing hierarchy. This insight complements prior work[8,9] which proposed that the progressive increase in receptive field size causes stronger simultaneous suppression in higher-level areas.

Increasing receptive field size and compression from early to higher-level visual areas have been interpreted as increasing summation windows that enhance invariance both in space[22,27,50,60,61] and time[28,32,33,37,39,40,44]. This aligns with the idea that spatial and temporal compression of visual information shares a similar processing strategy[40] and suggests that CST summation may be a general computational principle in the visual cortex. Moreover, our findings show that spatiotemporal receptive field models can be leveraged to gain insights about neural responses beyond the processing of visual motion and dynamic information, such as predicting responses to rapidly presented stimuli varying in spatial locations, as in the present study. In addition to predicting the level of simultaneous suppression, the CST pRF model showed that ventral visual areas are highly sensitive to the temporal properties of the visual input. These findings are in line with prior work showing that dynamic visual inputs affect not only motion-sensitive neurons in V1 and MT but also drive ventral visual stream areas V2, V3, hV4, and VO[36,62–68].

What may be the role of compressive spatiotemporal summation? Little is known regarding to the role of CST summation outside of motion processing[69–72]. One possibility is that increasing CST summation generates representations that encode complex shape and motion information that unfolds over time[73]. This may be useful for binding different views of novel objects during unsupervised learning associated with ventral stream functions[74,75] or for perceiving complex visual dynamics, actions, and social interactions associated with lateral stream functions[76–78]. Another possibility is that spatiotemporal compression within pRFs may enable neurons to prioritize novel visual information[13,79]. This may be beneficial for visual search[1,2] or short-term visual working memory by converting redundant visual information into a more efficient representation[80]. However, spatiotemporal compression may also limit visual processing capacity, affecting downstream cognitive processes such as worse memory for simultaneously vs sequentially presented items[81]. Thus, an important future direction is characterizing and computationally linking the neural phenomenon of simultaneous suppression to behaviors such as visual capacity, and testing what computational mechanisms generalize across scenarios and tasks.

In sum, our empirical data and voxel-wise pRF modeling approach call for a rethinking of the neural mechanisms that drive simultaneous suppression and suggest that suppression is a byproduct of compressive spatiotemporal computations. These findings provide an exciting new framework to computationally understand how stimulus content, context, and task demands affect simultaneous suppression and visual processing capacity more broadly.

## Methods
### Participants
Ten participants (six self-identified as female, four self-identified as male, ages 22–53 years, $M = 30.1$ years, SD = 8.7 years) with normal or corrected-to-normal vision participated in a retinotopy and SEQ-SIM fMRI experiment. Participants were recruited from the Stanford University community, including the three authors. Participants gave written informed consent, were compensated for their time, and all procedures were approved by the Stanford Internal Review Board on Human Subjects Research. We did not consider sex and/or gender in this study design.

### Stimuli and experimental design
Stimuli were generated using MATLAB R2017b (MathWorks, MA, USA) and PsychToolbox[82] on an Apple MacBook Pro laptop. Images were presented using an Eiki LC-WUL100L projector (Eiki International, Inc., CA, USA) on a rear-projection screen via two large mirrors placed at the back of the MRI scanner bed. The projected image had a resolution of 1920 × 1080 pixels, resulting in a field-of-view of ~38 × 24°, and a refresh rate of 60 Hz. The display was calibrated using a linearized lookup table.

**Retinotopy experiment.** Participants completed four 3.4-min runs, where bar stimuli cropped from colorful cartoons traversed across a 24 × 24° circular aperture (Toonotopy[56]). Cartoon images inside the bar changed randomly at 8 Hz. The bar swept in 12 discrete steps, 2-s per bar position, for four orientations (0°, 45°, 90°, and 135°) and two motion directions for each orientation. Observers fixated on a central dot (diameter = 0.12°) and pressed a button every time the fixation dot changed color (semi-random intervals, 6–36 s). Due to a coding error, button presses were only recorded for 3 participants, who performed at the ceiling ($M = 98.7\%$ correct, SD = 1.2%).

**SEQ-SIM experiment.** Participants completed eight ~5.5-min runs (except for participant S5, completing six runs), where eight squares were presented sequentially or simultaneously while fixating: four squares in the lower left quadrant and four squares in the upper right quadrant. Both sequential and simultaneous conditions used two presentation timings (short: 0.2 s and long: 1 s) and two sizes (small: 2° × 2° and big: 4° × 4°), resulting in eight conditions.

**Stimuli.** Squares were randomly cropped from colorful cartoons and placed on a mean luminance gray background. To ensure square stimuli would elicit responses in visual cortex, squares with little to no contrast were excluded (normalized root mean square contrast across pixels <10%). The content of individual squares differed for each trial and quadrant, and never repeated within a run. Within a quadrant, squares had a 2-by-2 layout with a 0.82° gap between them, centered at ~7.1° eccentricity ([$x,y$] = [5°,5°]). Both sizes used identical gap and eccentricity, such that four small squares extended horizontally and vertically from 2.59° to 7.41°, and big squares extended from 0.59° to 9.41°. The lower left and upper right quadrants had the same square locations but mirrored horizontally and vertically.

**Experimental design.** Stimuli were shown in ~8-s blocks, interspersed by 12-s blank periods. Each run started with a 6-s countdown and a 12-s blank and ended with a 12-s blank. Each condition was repeated four times in a pseudo-randomized order across two runs. The block order, as well as individual square presentation within a block, differed across runs. Each participant was assigned a unique pair of runs, which were repeated four times (three for participant S5) within the experiment with different square content (see example: https://osf.io/7rqf4).

Sequential and simultaneous conditions had eight trials per block for short stimuli and two trials per block for long stimuli. We used different trial-per-block ratios such that short and long conditions had a similar total block duration while the number of visual transients quadrupled (16 vs 64)—matching the increase between small and big square sizes (4 vs 16 deg²). In a sequential trial, the four squares in each quadrant appeared one at a time, in random order, with a 33-ms ISI between squares. In a simultaneous trial, all four squares in a quadrant appeared at once for the same duration and location followed by a

mean luminance gray display to match the duration of a sequential trial.

Block onsets and stimulus conditions were identical across quadrants, but the timing and order of individual square appearances were independently determined per quadrant. In simultaneous blocks with long stimulus presentations, stimuli in the first trial were presented at block onset to match sequential blocks. Stimuli of the second trial were presented 4 s later to avoid 7-s gaps between stimuli within a block. In simultaneous blocks with short presentations, stimuli in the first trial were also locked to block onset, but the onset of stimuli in the following seven trials was randomized within a trial.

**Task and behavioral performance.** Participants performed a 1-back letter RSVP task at fixation and pressed a button when a letter was repeated (1/9 probability). The letters (diameter of -0.5°) updated at 1.5 Hz, alternating between black and white colors, and randomly drawn from a predefined list ('A', 'S', 'D', 'F', 'G', 'H', 'J', 'K', 'B', and 'P'). Participants had a 0.83-s response window after a letter appeared and performance was displayed after every run. Outside the scanner, participants did 1-min practice runs until they reached at least 70% correct before starting the experiment. In the scanner, participants performed the task well ($M = 88\%$ correct, $SD = 8.2\%$), ranging from 68–95% correct, and an average false alarm rate of 2%. These behavioral data are confirmed by steady fixation in eye movement data (Supplementary Fig. 5) and indicate that participants were fixating throughout the experimental runs.

## MRI data acquisition

Participants' structural and functional data were collected using a 3 Tesla GE Signa MR750 scanner located in the Center for Cognitive and Neurobiological Imaging at Stanford University. Whole brain T1-weighted anatomy data were acquired using a BRAVO pulse sequence (1 mm$^3$ isotropic, inversion time = 450 ms, TE = 2.912 ms, FA = 12°), using a Nova 32-channel head coil. Functional data were collected using a Nova 16-channel coil, using a T2*-sensitive gradient echo planar imaging sequence (2.4 mm$^3$ isotropic, FoV = 192 mm, TE = 30 ms, FA = 62°). EPI slice prescriptions were oblique, roughly perpendicular to the calcarine sulcus. The retinotopy experiment used a TR of 2000 ms and 28 slices. SEQ-SIM experiment used a TR of 1000 ms and 14 slices. A T1-weighted inplane image (0.75 × 0.75 × 2.4 mm) was collected with the same coil and slice prescription as the functional scans to align functional and anatomical scans.

Left eye gaze data of nine participants were continuously recorded in each SEQ-SIM run at 1000 Hz using an EyeLink 1000 (SR Research Ltd., Osgoode, ON, Canada). Eye position calibration and validation were conducted before the first run, using a 5-point grid. We could not collect eye gaze data in one participant due to constraints in the mirror setup. Four participants were excluded prior to analysis due to excessive measurement noise. Analysis details for eye gaze data are in the Supplementary Material above Supplementary Fig. 5.

## MRI data analysis

**MRI preprocessing.** Whole-brain T1-weighted scans were aligned to the AC-PC line using SPM12 (https://github.com/spm/spm12) and auto-segmented with FreeSurfer's *recon-all* auto-segmentation[83] (v6.0; http://surfer.nmr.mgh.harvard.edu/). Small manual corrections of segmentations were executed with ITK SNAP (v3.6.0; http://www.itksnap.org/pmwiki/pmwiki.php). Functional data were slice-time corrected, motion corrected, drift corrected, and converted to percent signal change using the Vistasoft toolbox (https://github.com/vistalab/vistasoft). Participants' functional scans were aligned with the inplane to their whole brain anatomy scan, using a coarse, followed by a fine 3D rigid body alignment (6 DoF) using the alignvolumedata_auto toolbox (https://github.com/cvnlab/alignvolumedata). The first 8 (SEQ-SIM) or 6 (Retinotopy) volumes of each functional scan were removed to avoid data with unstable magnetization.

**Retinotopy analysis.** Retinotopy runs were averaged and analyzed with Vistasoft's CSS pRF model[27] using a 2-stage optimization (coarse grid-fit, followed by fine search-fit). For each voxel, this resulted in 2D Gaussian pRF with center coordinates ($x_0$, $y_0$) in degrees, pRF standard deviation ($\sigma$) in degrees, and pRF static nonlinearity exponent (CSS$_n$) ranging from 0.01 to 1. To avoid pRFs that are not visually responsive, we selected pRFs with $R^2 \geq 20\%$ in the retinotopy experiment, similar to previous pRF publications[56,84].

**Defining visual areas.** Spatial pRF parameters were converted to polar angle and eccentricity maps and projected to the participant's native cortical surface using nearest neighbor interpolation. Visual field maps were used to define the following visual areas: V1, V2, and V3[85], hV4 and VO1/2[86], LO1/2 and TO1/2[87], and V3A/B and IPS0/1[88].

**Defining ROIs and selecting voxels.** For each visual area, we selected voxels with pRFs centers within the circumference of the big squares in the SEQ-SIM experiment, that is, within an 8.82° × 8.82° square located 0.59°–9.41° from the display center in both $x$- and $y$-dimensions in each quadrant. From these voxels, we used those with corresponding data from the SEQ-SIM experiment. Overall, we obtained data in most participants' visual areas, except six participants who had insufficient coverage of IPS0/1 and two participants who had insufficient coverage of TO1/2, due to fewer slices in the SEQ-SIM experiment than in the retinotopy experiment.

**SEQ-SIM analysis.** We excluded voxels with a split-half reliability < 10% to filter out those voxels with little to no visual response. Excluded voxels were mostly from V1 and V2, with small pRFs that fell in between stimuli or on the border of stimuli. The two unique SEQ-SIM runs were concatenated for each repeat. When applying split-half cross-validation for model fitting, the four concatenated runs were split into two odd and two even runs, and averaged within each half.

Both observed and predicted run time series were averaged across split-halves and segmented into 23-TR time windows. These time windows spanned from 4 s pre-block onset, 8 s stimulus block, to 11 s post-block. For each voxel, we took the average time window and standard error of the mean (SEM) across four repeats. The average data and model time windows were summarized into eight values per voxel (one per condition), by averaging the BOLD response within a 9-TR window centered on the peak, spanning from either 4–12 s or 5–13 s after stimulus block onset. These values were used in the LMMs and scatter plots. We used a variable start per condition and visual area because the BOLD accumulation rate differed. The start was determined by averaging (data or model) time windows across voxels within a visual area and condition, into a 'grand mean' time window and finding the first TR after block onset where the BOLD response exceeded 10% of the total cumulative sum. This averaging window was applied to all voxels within a visual area.

## Linear Mixed Model

To quantify simultaneous suppression, we fitted a linear mixed model (LMM) to all participants' voxels within a visual area with MATLAB's *fitlme.m*, using the maximum likelihood fitting method. This LMM predicted the average simultaneous BOLD response of each voxel as a function of the average sequential BOLD response, for each stimulus condition (fixed interaction effect), allowing for a random intercept and slope per participant and stimulus condition (random interaction effect):

$$SIM\ ampl \sim 1 + SEQ\ ampl \times Condition$$
$$+ (1 + SEQ\ ampl \times Condition \mid Participant) \quad (1)$$

where SIM ampl and SEQ ampl are matrices (nr voxels × 4) with continuous values, Condition is a categorical vector (1 × 4), and Participant

is the group level for the random effects (ten participants for main pRF models (LSS, CSS, CST, and DoG), 7 participants for $CST_{fix}$, $CST_{op}$, DN-ST pRF models).

This LMM captured our data well (mean $R^2$ = 90%, SD = 6.6%), with V1: 86%, V2: 94%, V3: 94%, hV4: 92%, VO1/2: 97%, V3A/B: 95%, IPS0/1: 88%, LO1/2: 85%, and TO1/2: 76%). We tested this LMM against three alternative LMMs: (i) mean sequential amplitude as a fixed factor (no condition interaction effect) with one random intercept per participant, (ii) a fixed interaction effect with a single intercept per participant, identical for each stimulus condition, and (iii) a fixed interaction effect with a random participant intercept for each condition. Despite having more degrees of freedom (45) than the alternative LMMs (4, 10, and 19), the main LMM was a better fit to the data as it had a significantly higher log-likelihood than alternative LMMs, and lower AIC and BIC for each visual area (Supplementary Fig. 6 and Supplementary Table 5).

**Summarizing simultaneous suppression effects.** We summarized LMM results for each condition and visual area as line fits with $CI_{95\%}$ using the slope and intercept of the individual participants (Figs. 2b and 3b) or average across participants (Fig. 4a). For Fig. 4b, we summarized the simultaneous suppression level using the average slope and SEM across participants. For Fig. 8, we first averaged slopes across conditions within a participant, and then averaged slopes across participants (± SEM). For Supplementary Fig. 2a, we assessed differences in suppression levels between consecutive visual areas. We computed the difference in suppression levels separately for each participant by subtracting each participant's regression slopes of the earlier visual area from the subsequent visual area. Visual area pairs are chosen based on a feedforward visual hierarchy for each visual processing stream[76–78]. We then calculated the average difference in suppression across ten participants (± SEM) for each visual area pair and each stimulus condition. For Supplementary Fig. 2b, c, we first calculated the average slopes across stimulus conditions within a participant, then computed the difference in suppression slopes between pairs of visual areas within participants, and then computed the average difference in suppression across participants (± SEM).

### pRF modeling framework
Our modeling framework contained three main pRF models: (i) LSS, to test linear spatial summation[18], (ii) CSS, to test compressive spatial summation[27], and (iii) CST, to test compressive spatiotemporal summation[51]. Both LSS and CSS models linearly sum over the temporal duration of the stimulus.

Each model's input is a 3D binarized stimulus sequence, pixels by pixels (in visual degrees) by time (milliseconds). Each pRF is applied to each frame of the stimulus sequence to predict the neural pRF response. For each model, this neural response is then convolved with a canonical HRF (double-gamma SPM default) and downsampled to the fMRI acquisition TR. This results in a predicted BOLD response for the entire stimulus sequence. For each pRF that overlapped stimuli in the SEQ-SIM experiment, predictions were computed for each unique 5.5-min run, and then concatenated for the two unique runs. Importantly, concatenated runs contained all eight stimulus conditions, requiring each model to predict all conditions simultaneously.

We model spatial pRF parameters in each voxel using independent retinotopy data and then test which type of spatial and/or spatiotemporal pRF computations predict simultaneous suppression in each voxel in the main SEQ-SIM experiment.

**LSS pRF model.** The LSS model has a circular 2D Gaussian pRF with an area summing to 1. The pRF computes the dot product between the 2D Gaussian and stimulus sequence at each time point to predict the

response of the neural population within a voxel:

$$\text{Neural response}_{LSS}(t) = \int S(x,y,t) \cdot G(x,y)dxdy \quad (2)$$

$$G(x,y) = e^{\frac{(x-x_0)^2 + (y-y_0)^2}{2\sigma^2}} \quad (3)$$

where $t$ is time in ms, $S$ is the stimulus sequence with visual field positions $(x, y)$ in visual degrees by time in ms, and $G$ is a circular 2D Gaussian centered at visual field positions $(x_0, y_0)$ with size in standard deviation ($\sigma$) in visual degrees.

This model sums inputs linearly in visual space and time, and typically predicts the same BOLD response for sequential and simultaneous trials that are matched in stimulus size, location, and duration. For longer stimulus durations, the LSS model occasionally predicts larger responses for simultaneous than sequential conditions, due to a difference in ISI between the two conditions. Specifically, the randomized square onset causes sequential ISIs to range from 1–7 s, which by chance can be longer than the fixed 4-s simultaneous ISI—especially for small pRFs that overlap a single square. For these scenarios, the LSS model predicts the BOLD responses accumulate less in the sequential than simultaneous block.

**CSS pRF model.** The CSS model has the same spatial pRF model as the LSS model, followed by a static power-law nonlinearity:

$$\text{Neural response}_{CSS}(t) = \left[ \int S(x,y,t) \cdot G(x,y)dxdy \right]^{CSS_n} \quad (4)$$

where $t$ is time in ms, the power-law exponent ($CSS_n$) is bound between 0.01 and 1 and therefore results in compressive (subadditive) summation.

**CST pRF model.** The CST model contains three spatiotemporal channels for each voxel. Each channel has the same spatial pRF as the LSS model, which is combined with a sustained, on-transient, or off-transient neural temporal impulse response function (IRF).

For the main analysis, the sustained, on-transient, and off-transient IRFs are identical across voxels, where neural IRFs are based on the following gamma function:

$$IRF(t) = \frac{\left(\frac{t}{\kappa\tau}\right)^{(n-1)} e^{-\left(\frac{t}{\kappa\tau}\right)}}{\kappa\tau(n-1)!} \quad (5)$$

where $t$ is time in ms, $\kappa$ is the time constant ratio parameter, $\tau$ (tau) is the time constant parameter in ms, and $n$ is an exponent parameter.

The sustained IRF is a monophasic gamma function as in Eq. 5 with $\kappa = 1$, $\tau = 49.3$ ms and $n = 9$, resulting in a peak response between 40 ms and 50 ms. The on-transient IRF is the difference of two gamma functions: the sustained IRF and a second gamma function as in Eq. 5 with parameters: $\tau = 4.93$ ms, $\kappa = 1.33$, $n = 10$, resulting in a biphasic function that generates a brief response at stimulus onset. These IRF parameters are default V1 parameters from refs. [39,41], which are based on human psychophysics[89]. The off-transient IRF is identical to the on-transient IRF but with an opposite sign, generating a response at stimulus offset. The area under the sustained IRF sums to 1, and the area under each transient IRF sums to 0.

For each channel, we compute the spatiotemporal pRF response to a stimulus sequence by first applying the dot product between the spatial pRF and the stimulus sequence at each time point (ms resolution). This spatial pRF output is then convolved with the channel's neural temporal IRFs. The resulting spatiotemporal response is then rectified to remove negative values in the transient channels as we reasoned that either on- or off-transient responses will increase BOLD

responses (sustained responses are always positive). The rectified sustained, on-transient, and off-transient channel responses are then subject to the same static power-law nonlinearity, controlled by the exponent parameter ($CST_n$), resulting in the following neural response for each channel:

$$\text{Neural response}_{CST_i}(t) = \left[ \left| IRF_i(t) * \left[ \int S(x,y,t) \cdot G(x,y)dxdy \right] \right| \right]^{CST_n}$$

(6)

where $i = [1, 2, 3]$ indicates the sustained, on-transient, or off-transient channel, $t$ is time in ms, $*$ indicates convolution, and the power-law exponent ($CST_n$) is bound between 0.1 and 1, compressing the spatiotemporal channel responses. We use compressive nonlinearity as we reasoned that simultaneous suppression is due to subadditive summation.

After predicting neural responses for each channel, we sum on- and off-transient channels. The sustained channel and combined transient channels are then convolved with the HRF and downsampled to 1-s resolution to predict fMRI data. The voxel's response is the weighted sum of the two ($\beta_s$, $\beta_t$) time series. More details about the CST model can be found in ref. 51.

**Alternative pRF models.** In addition to the main pRF models, we tested two alternative spatiotemporal pRF models—$CST_{opt}$ and DN-ST —for 7 of our participants who took part in a separate spatiotemporal retinotopy experiment[51], and a difference of Gaussians (DoG) pRF model[48]. We used $CST_{opt}$ and DN-ST pRF model parameters that were optimized with a default HRF to match the HRF of the main CST pRF model).

**Optimized compressive spatiotemporal ($CST_{opt}$) pRF model.** The same model as the above-mentioned CST model, but using optimized spatial and temporal pRF parameters for each voxel, estimated from a separate spatiotemporal retinotopy experiment (see ref. 51), to understand the impact of using mostly fixed pRF parameters in the CST model. When describing $CST_{opt}$ model performance, we refer to the main CST model as $CST_{fix}$, because its pRF parameters are fixed (spatial parameters from the independent retinotopy experiment, a fixed time constant from refs. 39,41), except for the exponent ($CST_n$), which is estimated from the SEQ-SIM data.

**Delayed normalization spatiotemporal (DN-ST) pRF model.** A delayed divisive normalization spatiotemporal pRF model, to test if CST summation within the pRF with a different mathematical form can equally well predict the SEQ-SIM data. The DN-ST model implements subadditive summation using divisive normalization and an exponential decay function[40,42,44]. As for the $CST_{opt}$, DN-ST model parameters are optimized for each voxel from a separate spatiotemporal retinotopy experiment by Kim et al.[51].

The DN-ST model has a 2D circular Gaussian spatial pRF combined with a temporal IRF that contains a divisive normalization and an exponential decay function as implemented in ref. 51:

$$\text{Neural response}_{DN-ST}(t) = \frac{|r(t)|^n}{\sigma_{DN}{}^n + \left[ |r(t)| * IRF_{DN2}(t) \right]^n}$$

(7)

where $t$ is time in ms, $\sigma_{DN}$ is a semi-saturation constant, $n$ is the exponent, and $r(t)$ is the linear component of the neural response and computed as the convolution between the neural temporal $IRF_{DN1}$ and the spatial pRF response to the stimulus sequence (same as Eq. 2):

$$r(t) = IRF_{DN1}(t) * [S(x,y,t) \cdot G(x,y)]$$

(8)

where $IRF_{DN1}$ is a gamma function with time $t$ in ms and first time constant $\tau_1$ in ms:

$$IRF_{DN1}(t) = te^{-t/\tau_1}$$

(9)

The second temporal IRF in the denominator of the neural response acts as a low-pass filter using an exponential decay function:

$$IRF_{DN2}(t) = e^{-t/\tau_2}$$

(10)

where $t$ is time in ms and $\tau_2$ is the second time constant parameter in ms. More details about the DN-ST model, spatiotemporal retinotopy experiment, and optimization procedure can be found in ref. 51.

**Difference of Gaussians (DoG) pRF model.** To test whether center-surround DoG pRFs predict simultaneous suppression, we simulate each voxel's pRF as the difference between two 2D Gaussians, a center Gaussian from which a larger surround Gaussian is subtracted[48]. We simulate DoG pRFs only for V1–hV4 for three reasons. First, the surround pRF needs to encompass more visual input than the center pRF to predict simultaneous suppression. These visual areas have pRFs where the smaller pRF center will likely encompass a subset of square stimuli and the larger suppressive surround pRF will likely encompass more squares than the center. Second, the surround pRF of voxels in visual areas beyond hV4 will likely extend far beyond the visual display, unlikely to make a significant contribution to the response. Third, the effects of surround suppression are most prominent at early processing stages, including LGN[24] and early visual cortex (V1–V3[48,50]).

The center Gaussian is identical to the LSS pRF, estimated from the retinotopy session (Eq. 2). The surround Gaussian has the same center position with a larger size, where the scale factor was based on the average center/surround size ratio from ref. 50; V1: 7.4, V2: 6.8, V3: 7.3, and hV4: 5.8 times the center size. We used a constant scaling for all voxels within the same visual area, because directly estimating DoG pRFs from the independent retinotopy data using the approach by Zuiderbaan et al.[48] resulted in unstable model fits. This instability is likely due to the relatively few and short blank periods in our retinotopy experiment compared to Zuiderbaan et al., which has $4 \times 30$ s-blank periods for each 5.5-min run. We used scale factors by Aqil et al. as Zuiderbaan et al. do not report average center-surround scale factors within a visual area and data are limited to V1, V2, and V3. We did not implement the divisive normalization pRF model as described by Aqil et al.[50]

**Model fitting**

We fitted each voxel's pRF model prediction separately to data, using a split-half cross-validation procedure. The maximum height of predicted BOLD run time series was normalized to 1 and we added a column of 1's to capture response offset. This resulted in two regressors ($\beta_0$, $\beta_1$) for LSS and CSS models, and three regressors ($\beta_0$, $\beta_s$, $\beta_t$) for CST. We used linear regression (ordinary least squares) to fit these regressors to the voxel's observed run time series, separately for odd and even splits. To determine model goodness-of-fit (variance explained), we computed the cross-validated coefficient of determination (cv-$R^2$) by using the scaled predicted run time series of one split to predict observed run time series from the other split and vice versa (i.e., $\beta$-weights are fixed and not refitted). Cv-$R^2$ values and $\beta$-weights were averaged across split halves for each voxel. Split-half reliability across runs was used as the noise ceiling.

To check whether CST model performance could be inflated by the extra regressor, we also computed cross-validated adjusted-$R^2$, which penalizes goodness-of-fit for the number of time points and explanatory variables. The adjusted-$R^2$ values were almost numerically identical to $R^2$ and did not significantly affect our results or statistical comparisons.

**Fixed and optimized pRF parameters**. Spatial pRF parameters were independently estimated from each participant's retinotopy experiment using the CSS pRF model, resulting in a pRF center ($x_0$, $y_0$), standard deviation ($\sigma$), and exponent ($CSS_n$) parameter for each voxel. The standard deviation and exponent parameter trade-off in the CSS model (see ref. 27), where $\frac{\sigma}{\sqrt{CSS_n}}$ approximates the effective pRF size: the standard deviation ($\sigma$) estimated with a linear pRF model (LSS, no spatial compression). Therefore, to reconstruct CSS pRFs, we use each voxel's estimated CSS parameters ($x_0$, $y_0$, $\sigma$, and $CSS_n$). To reconstruct LSS and CST pRFs, we use the same estimated pRF center ($x_0$, $y_0$), but for the standard deviation ($\sigma$) we use the effective pRF size ($\frac{\sigma}{\sqrt{CSS_n}}$).

The time constant parameter ($\tau$) for the temporal IRF in the main CST model is fixed (from refs. 39,41) and we only optimized the exponent ($CST_n$) using a grid-fit approach for each voxel. The best fitting $CST_n$ was determined by systematically evaluating the goodness-of-fit of the predicted time series with $CST_n$ between 0.1 and 1 (0.05 steps) and selecting the $CST_n$ resulting in the highest cv-$R^2$. We used a grid-fit instead of a search-fit optimization approach to avoid estimates getting stuck in a local minimum.

We used a fixed temporal IRF for the following reasons. First, we predicted that the main driver of the suppression effect would be the compressive static nonlinearity (power-law exponent). Second, by using the same spatial parameters for all pRF models estimated from the independent retinotopy experiment, the model comparison will be more informative as differences in model performance are due to differences in nonlinear computations, not spatial position. Third, to estimate all CST parameters we need a separate spatiotemporal pRF retinotopy experiment[51]. We have such data for 7 out of ten participants, which we used for comparing $CST_{fix}$, $CST_{opt}$, and DN-ST spatiotemporal models. For this comparison, we restricted our analysis to voxels whose pRF centers overlap the square stimuli in the SEQ-SIM experiment, as well as voxels whose variance explained by the spatiotemporal pRF model was 20% or higher, and whose split-half reliability in the SEQ-SIM experiment was 10% or higher. As in the spatiotemporal retinotopy experiment, we excluded voxels with CST time constants ($\tau$) larger than 1000 ms. This sub-selection of voxels resulted in a substantially smaller number of voxels per visual area than we used to compare the main pRF models (~60% of the total) and resulted in the removal of IPS0/1 from our results as only two participants contributed to this visual area.

**Summarizing pRF parameters**

We resampled pRF size, exponents ($CSS_n$, $CST_n$, DN-ST $n$), time constants ($CST_{opt}$ $\tau$, DN-ST $\tau_1$ and $\tau_2$), DN-ST semi-saturation constants ($\sigma_{DN}$), and CST $\beta_s$ and $\beta_t$ 1000 times with replacement within a participant's visual area, because the number of voxels varied across areas and participants. For pRF size, exponents, time constants, and semi-saturation constant, we report the median resampled parameter for each participant and visual area because the V1 and V2 $CST_n$ were not normally distributed (see Supplementary Fig. 4). CST $\beta_s$ and $\beta_t$ were normally distributed; hence, we report the average resampled $\beta$-weights per participant and visual area. For group results, we report the average ($\pm$ SEM) across participants' mean or median resampled parameter value, for each visual area. For Supplementary Fig. 2b, c, we first compute the difference in median pRF size (or $CST_n$) for each visual area pair within participants, and then the average difference ($\Delta$, delta) in pRF size (or $CST_n$) values across participants ($\pm$ SEM).

**Statistical analyses**

To quantify differences in LMM regression slopes, we ran a two-way repeated measures ANOVA with factors of visual area and stimulus conditions across participants. To quantify differences in pRF model cv-$R^2$, we ran a two-way repeated measures ANOVA with factors pRF model and visual area (ROI) across voxels of all participants and visual areas. For both ANOVA results, if there was a main effect ($p < 0.05$), we used Bonferroni-corrected post-hoc multiple comparison $t$-tests (two-sided) to evaluate differences between pRF models, or visual area and stimulus condition. We used Pearson's correlation ($r$) to quantify the relationship between participant slopes averaged across conditions and effective pRF size, exponents ($CST_n$ or DN-ST $n$), time constants ($CST_{opt}$ $\tau$, DN-ST $\tau_1$ and $\tau_2$), or DN-ST semi-saturation constant ($\sigma_{DN}$) across visual areas.

## Reporting summary

Further information on research design is available in the Nature Portfolio Reporting Summary linked to this article.

## Data availability

The fMRI and behavioral data, model fits, and parameter data generated in this study have been deposited on the Open Science Framework (OSF) database[90] (https://osf.io/rpuhs/). The model fits and parameter data that relate to the Supplementary Information are available on OSF (https://osf.io/e83az/). All source data files and custom-written code necessary to reproduce the data figures in this paper are publicly available in the OSF database[90] https://osf.io/rpuhs/ and GitHub (see Code Availability). Source data are provided with this paper.

## Code availability

Analyses were conducted in MATLAB (R2020b). Code for analysis and reproducing data figures from minimally preprocessed data are publicly available on GitHub (https://github.com/VPNL/simseqPRF; v1.0.0: https://doi.org/10.5281/zenodo.12658143)[91] and (https://github.com/VPNL/spatiotemporalPRFs; v1.0.1: https://doi.org/10.5281/zenodo.12658232)[92]. In addition, we used the following publicly-available software: SPM 12 (https://github.com/spm/spm12; commit version 3085dac), FreeSurfer https://surfer.nmr.mgh.harvard.edu/, v6.0)[83], ITK SNAP (http://www.itksnap.org/pmwiki/pmwiki.php; v3.6.0), Vistasoft toolbox (https://github.com/vistalab/vistasoft; commit version 7f0102c), alignvolumedata toolbox (https://github.com/cnlab/alignvolumedata; commit version b513116). Identification by two-means clustering (I2MC) toolbox (https://github.com/royhessels/I2MC; commit version f39948d)[93].

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

## Acknowledgements

We thank Brian Wandell for fruitful discussions. This research was supported by the US NIH NEI R01 EY023915 (K.G.-S.). Funders had no role in study design, data collection and analysis, or decision to submit the manuscript.

## Author contributions

E.R.K. and K.G.-S. designed the experiment. K.G.-S. provided funding. E.R.K. and I.K. collected data and wrote the computational framework. E.R.K. analyzed the data. K.G.-S. oversaw the computational framework and data analysis. E.R.K. and K.G.-S. wrote and revised the manuscript. I.K. provided feedback on the initial and revised manuscript.

## Competing interests

The authors declare no competing interests.
