## [Peer Review File · Nature Communications]

REVIEWER COMMENTS

Reviewer #1 (Remarks to the Author):

In this paper the authors address the phenomenon of ‘simultaneous suppression’ : the observation that the response to two stimuli presented within a single receptive field is often less than the sum of the responses to the two stimuli presented separately.

What are the noteworthy results?

Through a careful set of voxel-level fMRI measurements and modeling they show that 1) SS is measurable in V1 and increases downstream from there. 2) A time-invariant compressive summation model captures much of the variance but a more sophisticated ‘CST’ model that accounts for fine-grained temporal effects is even more effective and 3) These models capture some of the effects that, in some branches of cognitive psychology, are more often attributed to ‘high level’ concepts such as biased competition.

Will the work be of significance to the field and related fields? How does it compare to the established literature?

This paper builds on several other recent works from this group and others that are attempting to build and validate realistic forward models of the visual system. For example, the models (CSS, CST) used here are described in other papers. However, the work is highly significant not so much for the approach (although the execution here is astonishingly thorough) but for the fact that it is applied to, and provides insight into an old and somewhat unexplained phenomenon in cognitive neuroscience.

Does the work support the conclusions and claims, or is additional evidence needed?

The work completely supports the conclusions with respect to the way that the temporal order of stimulus presentation (sequential or simultaneous) affects the amplitude of the resulting fMRI BOLD signal in different visual areas. I don’t think it supports a more general explanation of all simultaneous suppression phenomena - those listed in the introduction for example : “searching for a target among distractors, recognizing an object when surrounded by flankers, or keeping multiple items in short-term visual working memory.”. It’s hard to see how those could be due to reducing the apparent contrast of a (superthreshold) target.

Are there any flaws in the data analysis, interpretation and conclusions? Do these prohibit publication or require revision? Is the methodology sound? Does the work meet the expected

standards in your field? Is there enough detail provided in the methods for the work to be reproduced?

Methodologically this is a lovely paper - I appreciate the careful use of modeling and model testing, the thorough presentation of all the data, and the ability to compare different parametric models quantitatively. The work exceeds the normal standards in the field and there is great attention to data and code accessibility.

General points:

1: All the way through, I am reading about 'simultaneous suppression' and thinking 'but isn't this just surround suppression and/or contrast gain control'? At the end, the reveal is 'yes! - modern models of gain control more or less explain the whole thing and we don't have to think about the biased competition or 'dynamic allocation of cognitive resources' at least for fMRI response amplitude measures. But side-stepping this possibility right from the start and focusing on rather old 'cognitive' theories of simultaneous suppression makes for a frustrating read. By down-playing, for example, extraclassical suppressive receptive fields (e.g. Solomon et al 2002) or even just the known failure of response linearity when contrasts are added on top of each other (as far back as Boynton et al 1996 in fMRI) feels a little like attacking a 'straw man'. Am I missing something here? Is the presentation of two contrast stimuli within a single pRF not likely to engage standard multiplicative contrast gain control mechanisms?

2: Cumulative effects: The data are presented and discussed as though the suppression computations are executed independently in different areas. But in reality, the response in, say, VO1 must be inherited from earlier areas - and we know the suppression in those areas because we are measuring that as well. I realize that we do not know the exact manner in which signals are combined from area to area - but it might be helpful to at least mention this and even to plot the data in some cases as 'deltas' on earlier stages. For example, the story in Figure 4b is that a moderate amount of suppression is found in V2 - and far more than in V1. But in fact, since V2 (more or less) inherits its inputs from V1, the amount of suppression implemented in each of those two areas looks to be about the same (and by the same logic, V3 seems to do almost nothing).

Finally, I think it would be worth taking some time (perhaps in the supp mat) to describe the models (CSS and CST). I know they are in reference 41 but a) this is still an bioRxiv paper and b) they are so central to the current paper that it would be worth having them 'to hand' - along with the critical parameter values like the decay constant for the three different channels.

Minor issues: The word 'simulateneous' appears in figure captions 7 and 8.

Alex Wade

Reviewer #2 (Remarks to the Author):

Kupers et al apply a recently developed model to explain BOLD responses to spatiotemporally manipulated visual stimuli. I like the work, as it tries to explain a phenomenon (simultaneous suppression) that some might be tempted to think of as a high-level phenomenon as resulting from basic, low-level mechanisms.

The study is performed well, both in terms of analysis and in terms of experimentation. The writing is also high-level, although I think the explanatory figure 1 could use some work - but this is a matter of presentation and perhaps esthetics.

Below some of my points, ordered from conceptual to more specific.

Simultaneous suppression is not my specific focus, but I'm a bit puzzled by what this study adds to what we know about spatiotemporal nonlinearities in visual cortex, other than its narrative emphasis on simultaneous suppression. I think this is an interaction between the authors' narrative strategy and my understanding of the state of the field more than anything else, but perhaps the authors could add some statements to the text to clarify where they're coming from?

Given that both spatial and temporal compression are known to occur in human visual cortex, how reasonable is it to use models that only feature one of these to explain data in which spatial and temporal presentation properties are varied? I would consider these models valuable baselines, but not contenders. That means also that the CST model is the only serious contender to explain the data. I guess what I'm missing is a serious model comparison between different types of CST model that might be of theoretical interest. This point interacts with the impression that the manuscript leans very strongly on an as-yet unpublished modeling paper, which feels somewhat awkward at times; I sometimes wonder what it is that the present work does over and above that other paper - or is the present work -- in terms of modeling etc -- mostly a validation of the previously presented model?

The CST model has a very specific set of fixed timescales in its front-end whereas in the Groen study these timescale elements are parameters to be estimated, and which also vary across brain regions in an empirical sense. Why are these fixed in the present implementation? How would this possibly impact the results presented here?

One other question about the model is that the present CST model has a fixed power-law exponentiation analogous to the CSS model. However, with the Groen and earlier Zhou studies showing an explicit divisive normalization pool analogous to the explicit spatial normalization pool of the recent Aqil et al study, would a combination of these explicit temporal and spatial models not be a more appropriate modeling strategy? I don't mean to insinuate that the authors should perform this modeling, but I hope they could at least discuss this in more detail.

the explanatory figure 1 is somewhat confusing, and in my mind needs work. Mostly, much of the information is repeatedly depicted and it's unclear why that is. So as a reader, I end up looking for reasons why things are shown 2 or even 3 times - rather puzzling.

- only 2 conditions are shown in A, and then repeated in B, but for a reader it's unclear why these specific 2 conditions are repeated in B.

- In B, everything about the timings is displayed twice when the difference between the top and bottom subpanels is just spatial size.

- there is no visual indication how the 8s trials are embedded in the experimental run, but signal timecourses in figure 2 are from -5 to 15 s. I end up wondering about the blanks in between, something I have to look up in the methods.

REVIEWER COMMENTS

We thank both reviewers for their enthusiasm and valuable feedback. We addressed all the comments, see point-by-point answers in blue below. We believe that these changes have strengthened our manuscript.

Reviewer #1 (Remarks to the Author):

In this paper the authors address the phenomenon of ‘simultaneous suppression’: the observation that the response to two stimuli presented within a single receptive field is often less than the sum of the responses to the two stimuli presented separately.

a. What are the noteworthy results?

Through a careful set of voxel-level fMRI measurements and modeling they show that 1) SS is measurable in V1 and increases downstream from there. 2) A time-invariant compressive summation model captures much of the variance but a more sophisticated ‘CST’ model that accounts for fine-grained temporal effects is even more effective and 3) These models capture some of the effects that, in some branches of cognitive psychology, are more often attributed to ‘high level’ concepts such as biased competition.

b. Will the work be of significance to the field and related fields? How does it compare to the established literature?

This paper builds on several other recent works from this group and others that are attempting to build and validate realistic forward models of the visual system. For example, the models (CSS, CST) used here are described in other papers. However, the work is highly significant not so much for the approach (although the execution here is astonishingly thorough) but for the fact that it is applied to, and provides insight into an old and somewhat unexplained phenomenon in cognitive neuroscience.

c. Does the work support the conclusions and claims, or is additional evidence needed?

The work completely supports the conclusions with respect to the way that the temporal order of stimulus presentation (sequential or simultaneous) affects the amplitude of the resulting fMRI BOLD signal in different visual areas. I don’t think it supports a more general explanation of all simultaneous suppression phenomena - those listed in the introduction for example : “searching for a target among distractors, recognizing an object when surrounded by flankers, or keeping multiple items in short-term visual working memory.”. It’s hard to see how those could be due to reducing the apparent contrast of a (superthreshold) target.

d. Are there any flaws in the data analysis, interpretation and conclusions? Do these prohibit publication or require revision? Is the methodology sound? Does the work meet the expected standards in your field? Is there enough detail provided in the methods for the work to be reproduced?

Methodologically this is a lovely paper - I appreciate the careful use of modeling and model testing, the thorough presentation of all the data, and the ability to compare different parametric models quantitatively. The work exceeds the normal standards in the field and there is great attention to data and code accessibility.

Response to Reviewer 1’s point (c): Thank you for your clear and enthusiastic summary of the work, its quality, and its impact. With respect to your comment (c): Our rationale was to provide a scholarly background and mention several general examples that have been linked in prior literature to simultaneous suppression. We did not intend to argue that our work support supports all cases of visual

capacity limits. To clarify this, we have updated the text to separate the general behavioral phenomena (in the first paragraph) from the specific neural phenomenon we are testing here in the Introduction (new second paragraph) [pp. 3, starting line 20] and Discussion [pp. 26, line 617-619]: “Thus, an important future direction is characterizing and computationally linking the neural phenomenon of simultaneous suppression to behaviors such as visual capacity, and testing what computational mechanisms generalize across scenarios and tasks.”

General points:

R1.1: All the way through, I am reading about ‘simultaneous suppression’ and thinking ‘but isn’t this just surround suppression and/or contrast gain control’? At the end, the reveal is ‘yes! - modern models of gain control more or less explain the whole thing and we don’t have to think about the biased competition or ‘dynamic allocation of cognitive resources’ at least for fMRI response amplitude measures. But side-stepping this possibility right from the start and focusing on rather old ‘cognitive’ theories of simultaneous suppression makes for a frustrating read. By down-playing, for example, extraclassical suppressive receptive fields (e.g. Solomon et al 2002) or even just the known failure of response linearity when contrasts are added on top of each other (as far back as Boynton et al 1996 in fMRI) feels a little like attacking a ‘straw man’. Am I missing something here? Is the presentation of two contrast stimuli within a single pRF not likely to engage standard multiplicative contrast gain control mechanisms?

The Reviewer makes multiple points to which we respond separately:

Response 1.1A: Thank you for raising this point and asking why we focus on ‘old cognitive’ theories. We start from the biased competition theory because it is a very prominent framework to study simultaneous suppression. The theory is one of the most cited explanations in the field when simultaneous suppression occurs and why it depends on the allocation of attention (Desimone & Duncan 1995: 10,490+ times, Kastner & Ungerleider 2000: 2780+ times, according to Google Scholar in March 2024). Because the biased competition theory is a popular framework, simultaneous suppression is often explained in terms of a lack of allocated neural resources to process multiple objects in the visual field (and how it can be resolved by visual attention reallocating neural resources to behaviorally-relevant stimuli). So, in terms of scholarship, we believe the prevailing hypothesis of simultaneous suppression is the theory of biased competition.

Nonetheless, we agree with the Reviewer’s perspective. In fact, the starting point for the study was that it was not clear to us why the field had settled in a complex cognitive theory to explain simultaneous suppression before formally testing the role of bottom-up computations. The whole point of the present study is to provide a fresh perspective on an old enigma using a computational approach to test bottom-up visual processing as an explanation. While this approach seems sensible to this Reviewer, we would like to underscore that this has not been attempted before.

Action 1.1A: As we did not intend to downplay prior work on response nonlinearity in fMRI BOLD measurements and we agree with the Reviewer that we can provide context on previously observed BOLD nonlinearities earlier as it informed our current stimulus-driven approach. To address this, we have extended our Introduction to cite this work [pp. 4, second paragraph, lines 54-62]: “Although linear summation within receptive fields is observed in some cases²¹⁻²³, violations of response linearity in the human visual system have been extensively reported. Spatially, responses to bigger stimuli are typically less than the sum of responses to smaller stimuli^{22,24-27}. Temporally, responses to longer stimuli are typically smaller than the sum of responses to shorter stimuli²⁸⁻⁴⁴. While both hemodynamic⁴⁵⁻⁴⁷ and neural^{27,35,39-42,48-51} mechanisms may contribute to observations made with fMRI, empirical and

computational modeling studies suggest that the observed subadditivity is driven by compressive summation of visual inputs within neurons' receptive fields across space^{27,50} and across time^{33,36,38,40-44}."

Response 1.1B: The Reviewer comments that all "modern models of gain control more or less explain the whole thing". We respectfully disagree. While many observed neural and BOLD nonlinearities show subadditive summation for stimuli increasing in spatial extent, timing, or contrast, it is unclear whether all these observed phenomena can be explained by "more or less" the same model. This is because different subadditive models implement different operations and have been tested on limited stimulus sets or specific experimental designs. For example, in our study, the compressive spatial summation (CSS) model is able to capture a substantial amount of the suppression effects in 1-s stimulus duration conditions, but fails to predict the responses for conditions with brief stimulus durations (0.2-s). Had we not tested responses for brief stimuli, we may have concluded that the CSS model performance is not that far off in predicting simultaneous suppression, and in some visual areas it performs similarly to the CST model.

To test the Reviewer's hypothesis, and verify that simultaneous suppression is not just 'surround suppression and/or contrast gain control' but rather generated from compressive spatiotemporal computations by pRFs, we added analyses of two additional pRF models:

(i) Center-surround Difference of Gaussians (DoG) pRFs. The Reviewer refers to Solomon et al. (2002) and suggests center-surround suppression as a potential candidate mechanism that may generate simultaneous suppression. As center-surround models have successfully captured subadditivity of neural activity in early visual processing stages (retina, LGN, V1-V3), we agree with the Reviewer that it would be interesting to test if such a model can explain simultaneous suppression. Specifically, we tested if a DoG pRF model predicts simultaneous suppression in V1-hV4. We implemented DoG pRFs for these visual areas as their pRFs are (on average) small enough such that their center will cover some stimuli and their surrounds will cover other stimuli (for later regions stimuli fall with the DoG center and thus the surround could not contribute to simultaneous suppression). We modeled the center portion of the DoG pRFs' spatial parameters using the LSS spatial pRF and the surround portion of DoG pRFs as having same center position, but larger size. We used a fixed center/surround scale factor per visual area based on data published by Aqil et al. 2021 PNAS, instead of estimating DoG parameters directly from the retinotopy data because fits were not stable. That is, using the pRF model by Zuiderbaan et al. 2012 JoV resulted in unrealistically large surround sizes for relatively small centers. This instability is likely due to the relatively few and short blank periods in our retinotopy experiment compared to Zuiderbaan et al., which is designed to map DoG and has 4 x 30s-blank periods per 5.5-min run. Our results show that the DoG pRF model predicts similar responses for SEQ and SIM paired conditions, because these pRFs sum linearly over space and time. In other words, because paired SEQ/SIM conditions are matched in overall stimulus duration and size, DoG pRFs do not predict simultaneous suppression (**Supplementary Fig 7A,C**). Across voxels, the DoG model has similar or lower cross-validated variance explained ($cv-R^2$) than the LSS pRF model in all 4 tested visual areas (V1-hV4), which is worse than both CSS and CST models (**Supplementary Fig 7B**).

Action 1.1B: We have now simulated a center-surround DoG pRF model and show that this model doesn't predict simultaneous suppression. Results are reported in new **Supplementary Fig 7 [p. 53]** and on **p. 19** (third paragraph, line 413-423); The implementation is in the Methods [**p. 34, lines 849-866**].

B) Delayed Normalization Spatiotemporal pRFs. We also tested whether another form of compressive spatiotemporal summation can explain simultaneous suppression: a Delayed Normalization Spatiotemporal pRF model (DN-ST, Kim et al. 2024 J Neurosci), which temporal impulse response function is based on a temporal pRF model from Winawer and colleagues (Zhou et al. 2019 PLoS CB,

Groen et al. 2022 J Neurosci). We chose this model as this is a different type of spatiotemporal pRF model with subadditive properties, and we hypothesized that this may provide an alternative computational mechanism. As our current retinotopy experiment has only one temporal condition, it is insufficient for estimating DN-ST pRF model parameters. Thus, we used independently estimated DN-ST pRF parameters in each voxel from the spatiotemporal mapping experiment (Kim et al. 2024 J Neurosci) for the 7 subjects that participated in both studies. We find that, like the CST model, the DN-ST model predicts simultaneous suppression across spatial and temporal manipulations (new **Supplementary Figs 8 & 9, and Supplementary Table 4 [pp. 54-57]**). We find that the DN-ST model captures increasing levels of simultaneous suppression up the visual hierarchy, and varying suppression levels across stimulus manipulations (median $cv-R^2$ ranging from 7% to 34%). A two-way repeated measures ANOVA revealed significant effects of pRF model ($F(2)=1.1 \times 10^2$, $p=6.3 \times 10^{-47}$) and ROI ($F(7)=1.3 \times 10^3$, $p < 10^{-47}$) on $cv-R^2$, where the DN-ST pRFs predicts overall about 3% less $cv-R^2$ than CST pRFs, as well as a significant interaction between pRF model and ROI ($F(2,7)=5.0$, $p=1.6 \times 10^{-9}$) (**Supplementary Fig 8B and Supplementary Table 4**). When examining the model-based suppression levels, we find that DN-ST has a different pattern of errors: DN-ST pRFs well-predict responses to long stimuli (especially for small stimulus sizes) and predict less simultaneous suppression than observed for short duration stimulus conditions (new **Supplementary Fig 9**). The implementation of the DN-ST model is described in the Methods [pp. 33-34].

Action 1.1C: We tested a DN-ST pRF model [new **Supplementary Figs 8-9, and Table 4, pp. 54-57**]. We describe the results on p. 19-20 (lines 424-439), and model implementation on **Methods pp. 33-34**].

Action 1.1D: Lastly, we also now elaborate why we choose to test this subset of models to test simultaneous suppression effects [on Results p. 15, lines 315-322]:

“We test these pRF models for four main reasons. First, they describe a neural mechanism with a receptive field restricted to part of the visual field; this restriction is needed to test the impact of stimulus location and size. Second, the identical spatial pRF across models and similar static nonlinearity implementation for compressive models (CST and CSS) allow for informative comparisons between models. Third, both compressive models have the potential to predict simultaneous suppression within this stimulus regime. Fourth, CST and CSS models have been successful in providing a comprehensive explanation for subadditive visually-driven responses across visual cortex^{27,51}.”

R1.2: Cumulative effects: The data are presented and discussed as though the suppression computations are executed independently in different areas. But in reality, the response in, say, VO1 must be inherited from earlier areas - and we know the suppression in those areas because we are measuring that as well. I realize that we do not know the exact manner in which signals are combined from area to area - but it might be helpful to at least mention this and even to plot the data in some cases as ‘deltas’ on earlier stages. For example, the story in Figure 4b is that a moderate amount of suppression is found in V2 - and far more than in V1. But in fact, since V2 (more or less) inherits its inputs from V1, the amount of suppression implemented in each of those two areas looks to be about the same (and by the same logic, V3 seems to do almost nothing).

Response 1.2: Thanks for this comment, this is a good point. As larger receptive fields are thought to pool over smaller receptive fields with similar visual field positions when moving up the visual hierarchy, it is possible that suppression is accumulated by downstream areas throughout the visual pathways. One possibility is that suppression effects may increase monotonically across the visual hierarchy. An alternative prediction is that suppression increases initially and then plateaus once the receptive fields are large enough to contain all four squares.

To gain insight into the potential accumulation of suppression effects, we followed the Reviewer's recommendation to visualize the difference ("deltas") in average suppression slopes that follow the hierarchy in visual processing pathways in a new analysis (new **Supplementary Fig 2, p. 45**). We subtract in each participant the suppression level (LMM slope) between consecutive visual areas V2-V1 and V3-V2 (early), LO-V3 and TO-LO (lateral), hV4-V3 and VO-hV4 (ventral), and V3A/B-V3 and IPS0/1-V3A/B (dorsal). Our data show a more complicated story than the two predictions above, which we describe on **p. 13**. In short, differences in suppression levels across consecutive visual areas along visual pathways vary by stimulus condition (as expected from the main results), and do not increase consistently across the visual hierarchy nor plateau. In other words, we find that larger suppression in higher-level areas is not a sum of suppression in the preceding areas. We also do not find a clear relationship between the differences in suppressions levels across areas averaged across stimulus conditions and differences in pRF size or spatiotemporal compression (CST exponent) within the pRF. Together, this suggests that simultaneous suppression may accumulate to some extent from area to area due to increase pRF sizes from earlier to later visual areas, but it cannot fully be explained by accumulation alone.

Action 1.2: We added a new analysis examining the cumulative effect of suppression levels across consecutive visual areas in new **Supplementary Fig 2 [p. 45]** and describe the findings in the Results section [**p. 13**, second paragraph, lines 254-267].

R1.3: Finally, I think it would be worth taking some time (perhaps in the supp mat) to describe the models (CSS and CST). I know they are in reference 41 but a) this is still an bioRxiv paper and b) they are so central to the current paper that it would be worth having them 'to hand' - along with the critical parameter values like the decay constant for the three different channels.

Response 1.3: We aimed for conciseness, but we agree with the Reviewer that it is worth having more details about the models, so we extended the Methods to provide this information. The paper mentioned, in what was reference 41 (now reference 51), is now published in the Journal of Neuroscience. We have updated the reference: <https://www.jneurosci.org/content/44/2/e0803232023>.

Action 1.3: We have expanded the description and formulas of the various pRF models (LSS, CSS, CST, DN-ST, DoG) in the Methods [**pp. 31-34**].

R1.4: Minor issues: The word 'simulataneous' appears in figure captions 7 and 8.

Response 1.4: Thank you for catching!

Action 1.4: We fixed the typos.

Alex Wade

Reviewer #2 (Remarks to the Author):

Kupers et al apply a recently developed model to explain BOLD responses to spatiotemporally manipulated visual stimuli. I like the work, as it tries to explain a phenomenon (simultaneous suppression) that some might be tempted to think of as a high-level phenomenon as resulting from basic, low-level mechanisms.

The study is performed well, both in terms of analysis and in terms of experimentation. The writing is also high-level, although I think the explanatory figure 1 could use some work - but this is a matter of presentation and perhaps esthetics.

Thank you for your enthusiastic and concise summary of our work.

Below some of my points, ordered from conceptual to more specific.

R2.1: Simultaneous suppression is not my specific focus, but I'm a bit puzzled by what this study adds to what we know about spatiotemporal nonlinearities in visual cortex, other than its narrative emphasis on simultaneous suppression. I think this is an interaction between the authors' narrative strategy and my understanding of the state of the field more than anything else, but perhaps the authors could add some statements to the text to clarify where they're coming from?

Response 2.1: We agree with the Reviewer that we can clarify why we choose this narrative and discuss earlier how our study relates to prior work on nonlinearities in visual cortex. We believe that Reviewer 1 made a similar point (see response to point R1.1).

In brief, this study is not about general spatiotemporal nonlinearities in visual cortex, but specifically about understanding simultaneous suppression (as reflected in the title). We study simultaneous suppression because it is a prevalent phenomenon that has been studied for ~30 years and is the basis of the very influential biased competition theory of visual attention (Desimone & Duncan 1995: 10,490+ times, Kastner & Ungerleider 2000: 2780+ times, according to Google Scholar in March 2024). Yet, its underlying neural mechanisms remain unclear. The starting point for our study was the puzzlement of why the explanation of stimuli competing for resources within receptive fields and top-down mechanisms have been the predominant explanation of simultaneous suppression, before considering nonlinear bottom-up visual processing. This prompted us to examine bottom-up visual computations as a potential explanation for simultaneous suppression. We underscore that neither formal bottom-up computations nor spatiotemporal nonlinearities in the visual system have been tested to explain simultaneous suppression. These are new ideas in our study.

Additionally, while computational models of compressive spatial nonlinearities within receptive fields across visual cortex have been published within the last decade (e.g., Kay et al. 2013), such receptive field models of compressive temporal nonlinearities have only recently been considered (Zhou 2018 J Neurosci; Zhou 2019 PLoS CB; Stigliani 2019 PLoS CB; Groen 2022 J Neurosci).

In fact, there have been no measurements of compressive spatiotemporal nonlinearities in the human visual system until our very recent study (Kim et al., 2024 J Neurosci), as spatiotemporal receptive fields have only been examined with electrophysiology in 2 visual areas of macaque monkey: V1 and MT (e.g., DeValois et al. 1998 PNAS, DeValois et al. 2000 Vision Res; or Rust et al. 2005 Neuron). Thus, considering of the role of spatiotemporal processing by population receptive fields in the visual system and how these computations may contribute to simultaneous suppression—what is considered a cognitive phenomenon—is extremely novel. We now elaborate more on this motivation and also extended the Introduction to discuss previous studies that examined both neural and BOLD nonlinearities in the human visual system,

Action 2.1: We now mention prior work on observed nonlinearities in visual cortex in the Introduction [**pp. 4, second paragraph, lines 54-62**]: *“Although linear summation within receptive fields is observed in some cases²¹⁻²³, violations of response linearity in the human visual system have been extensively*

reported. Spatially, responses to bigger stimuli are typically less than the sum of responses to smaller stimuli^{22,24-27}. Temporally, responses to longer stimuli are typically smaller than the sum of responses to shorter stimuli²⁸⁻⁴⁴. While both hemodynamic⁴⁵⁻⁴⁷ and neural^{27,35,39-42,48-51} mechanisms may contribute to observations made with fMRI, empirical and computational modeling studies suggest that the observed subadditivity is driven by compressive summation of visual inputs within neurons' receptive fields across space^{27,50} and across time^{33,36,38,40-44}.

And how this relates to our choice of pRF model [p. 15, lines 315-322]: *"We test these pRF models for four main reasons. First, they describe a neural mechanism with a receptive field restricted to part of the visual field; this restriction is needed to test the impact of stimulus location and size. Second, the identical spatial pRF across models and similar static nonlinearity implementation for compressive models (CST and CSS) allow for informative comparisons between models. Third, both compressive models have the potential to predict simultaneous suppression within this stimulus regime. Fourth, CST and CSS models have been successful in providing a comprehensive explanation for subadditive visually-driven responses across visual cortex^{27,51}."*

R2.2: Given that both spatial and temporal compression are known to occur in human visual cortex, how reasonable is it to use models that only feature one of these to explain data in which spatial and temporal presentation properties are varied? I would consider these models valuable baselines, but not contenders. That means also that the CST model is the only serious contender to explain the data. I guess what I'm missing is a serious model comparison between different types of CST model that might be of theoretical interest. This point interacts with the impression that the manuscript leans very strongly on an as-yet unpublished modeling paper, which feels somewhat awkward at times; I sometimes wonder what it is that the present work does over and above that other paper - or is the present work -- in terms of modeling etc -- mostly a validation of the previously presented model?

Response 2.2A: While compressive temporal summation alone cannot explain simultaneous suppression—as suppression requires a spatial receptive field—spatial compression alone could have explained the phenomenon. Therefore, we respectfully disagree that the compressive spatial summation model (CSS) is a strawman. In fact, the idea that simultaneous suppression is due to stimuli competing when they fall within the spatial extent of the receptive field is so dominant in the field, that testing how systematically varying timing of stimuli may affect simultaneous suppression has not been considered until our current study. So, our study is novel not only in the computational modeling approach but also in its design and empirical observations to generate neural data that is necessary for distilling the underlying mechanisms. Indeed, our results show that the CSS model can predict simultaneous suppression for longer stimulus presentation timing (1-s) in several visual areas that have large pRFs (e.g., VO1/2), but typically fails to predict suppression for shorter stimulus presentations (0.2-s).

With respect to the baseline model, we agree that the linear spatial summation (LSS) is the baseline model. We included it here as prior papers reported no simultaneous suppression in V1, which is the predicted response from a pRF model that sums linearly in space and time such as the LSS model.

We agree that comparing the CST model to another compressive spatiotemporal model is theoretically interesting. Therefore, we followed on the Reviewer's suggestion and we added a new analysis examining how well a spatiotemporal pRF model with delayed normalization (DN-ST; Kim et al. 2024 J Neurosci) predicts simultaneous suppression. The DN-ST model has a spatial pRF that is a 2D circular Gaussian and its outputs also depend on temporally-delayed divisive normalization using an exponential decay function. We find that the performance of the DN-ST pRF model is similar to the CST model (thus, better than the CSS or LSS), with the overall cross-validated model performance being slightly lower (about 3%) than for the CST model (**Supplementary Fig 8**). A two-way repeated measures ANOVA with

revealed that this difference was significant, with main effects of pRF model ($F(2)=1.1 \times 10^2$, $p=6.3 \times 10^{-47}$) and ROI ($F(7)=1.3 \times 10^3$, $p < 10^{-47}$) on $cv-R^2$, as well as a significant interaction between pRF model and ROI ($F(2,7)=5.0$, $p=1.6 \times 10^{-9}$). Post-hoc Bonferroni-corrected t-tests show that both CST models significantly explain more $cv-R^2$ than the DN-ST model in all visual areas (except for CST_{opt} in TO1/2, see **Supplementary Table 4**). Similar to the CST model, the DN-ST model predicts simultaneous suppression that increases across the visual hierarchy and varies with stimulus condition. Interestingly, the DN-ST model captures stronger suppression levels up the visual hierarchy and stronger suppression levels for long (1-s) than short (0.2-s) stimulus timing, but tends to underpredict the observed suppression for the short stimulus durations (**Supplementary Fig 9A**).

Action 2.2A: To understand how other spatiotemporal models compare to the CST model, we have now implemented the Delayed Normalization Spatiotemporal pRF model (DN-ST) to test if it predicts simultaneous suppression for the 7 subjects that overlap with the Kim et al. study. Results are discussed on **p. 19-20** (last paragraph, lines 424-439), shown in **Supplementary Figs 8-9, and Table 4 [pp. 54-57]**, and the model is described in Methods **[pp. 33-34]**.

Response 2.2B: In response to the Reviewer's question what our study adds to the related preprint from our lab. We note that since our submission, the paper has been published in the Journal of Neuroscience: <https://www.jneurosci.org/content/44/2/e0803232023> (now ref 51). In Kim et al. 2024 J Neurosci, we developed a new framework to measure and estimate spatiotemporal pRFs from fMRI data and show how spatiotemporal pRF properties vary across visual cortex. However, beyond characterizing properties of spatiotemporal pRFs, Kim et al. 2024 does not use spatiotemporal pRFs to predict visual responses to other stimulus sequences or examines how differences in spatiotemporal pRFs across cortex may affect other visual processing. Here, we leveraged the opportunity afforded by the compressive spatiotemporal pRF model to operationalize and test what kinds of computations may generate simultaneous suppression at the voxel level. We show for the first time that spatiotemporal computations play a key role in processing rapidly changing static visual inputs (not just local motion stimuli), and that voxels in the human ventral visual stream are sensitive to stimulus timing (not just V1, MT, or TO in the lateral visual stream).

Action 2.2B: We expanded a Discussion paragraph to emphasize what our study contributes over the Kim et al. study **[p. 25, second paragraph, lines 598-605]**: *"Moreover, our findings show that spatiotemporal receptive field models can be leveraged to gain insights about neural responses beyond processing of visual motion and dynamic information, such as predicting responses to rapidly presented stimuli varying in spatial locations, as in the present study. In addition to predicting the level of simultaneous suppression, the compressive spatiotemporal pRF model showed that ventral visual areas are highly sensitive to temporal properties of the visual input. These findings are in line with prior work showing that dynamic visual inputs affect not only motion-sensitive neurons in V1 and MT but also drive ventral visual stream areas V2, V3, hV4, and VO^{36,62-68}."*

R2.3: The CST model has a very specific set of fixed timescales in its front-end whereas in the Groen study these timescale elements are parameters to be estimated, and which also vary across brain regions in an empirical sense. Why are these fixed in the present implementation? How would this possibly impact the results presented here?

Response 2.3: We agree with the Reviewer that using fixed time scales is a simplification of our CST model implementation. We decided to use fixed parameters for three main reasons. First, we use the same spatial parameters for all pRF models for an informative model comparison, estimated from the

independent retinotopy experiment, as differences in model performance are due to differences in nonlinear computations, not spatial position. Second, estimating spatial and temporal CST pRFs parameters in each voxel requires manipulating temporal and spatial parameters of the stimuli. Because our retinotopic mapping experiment uses the same image rate for each bar stimulus, we could not use these data to optimally estimate CST pRFs parameters for all our participants, in particular the time constant parameter (see also Kim et al. 2024). Thus, we used a fixed time constant based on psychophysical experiments (Watson 1986) and which we used previously to successfully predicted fMRI responses to time varying stimuli (Stigliani et al. 2017; 2019). Third, different participants participated in the present study and Kim et al. 2024. As such, we only have 7 overlapping subjects for which we can use the optimized CST parameters fitted to the spatiotemporal mapping experiment from Kim et al. 2024.

To understand how fixed temporal parameters impact our results, we performed a new analysis for the 7 participants who also participated in Kim et al. (2024) where we use CST pRFs with optimized parameters to predict responses in the SEQ-SIM experiment, and compared the prediction to the CST pRF model with a fixed temporal parameter. Since we used a canonical HRF for all voxels and models, we used Kim et al.'s spatiotemporal CST parameters (x_0 , y_0 , σ , τ , power law exponent) for each voxel estimated with a fixed HRF, rather than using a voxel-wise optimized HRF. Overall, we find that using optimized CST parameters results in similar cross-validated variance explained ($cv-R^2$) of SEQ-SIM data as using the CST pRF model with a fixed time constant (**Supplementary Fig 8**). In terms of predicting simultaneous suppression, the CST-optimized pRF model tends to predict less suppression for short stimulus timings than the CST-fixed pRF model (**Supplementary Fig 9A**). However, we find that the CST_{fix} model explains slightly, but significantly more $cv-R^2$ than the CST_{opt} in several visual areas (V1, hV4, V3A/B, LO1/2, and TO1/2), **Supplementary Table 4**). Together, these results suggest that: (i) across all visual areas and experimental conditions, independently estimated CST parameters for each voxel result in slightly more accurate predictions of simultaneous suppression (but can result in slightly but significantly less $cv-R^2$ on an area by area basis), and (ii) spatial and temporal pRF parameters interact, where different combinations of pRF parameters within a voxel can result in similar model performance in this experiment.

Action 2.3A: We now elaborate why we use fixed temporal pRF parameters for the CST model in the Methods [p. 35, lines 883-890].

Action 2.3B: We now tested the effect of optimized pRF parameters in the CST model for 7 participants overlapping with the Kim et al. study. Results are shown in **Supplementary Figure 8 and 9 [pp. 54-57]**, Methods are described on [pp. 33-35], and results are discussed on [p. 18, lines 392-393]:

*“The CST model perform similarly with pRF parameters that are optimized using the independent spatiotemporal retinotopy experiment⁵¹ (**Supplementary Figs 8-9**).”*

And [p. 21, lines 429-439]: *“A two-way repeated measures ANOVA reveals significant effects of pRF model ($F(2)=1.1 \times 10^{102}$, $p=6.3 \times 10^{-47}$) and ROI ($F(7)=1.3 \times 10^3$, $p < 10^{-47}$) on $cv-R^2$ (**Supplementary Fig 8B**), where the DN-ST model with optimized parameters predict overall less $cv-R^2$ than either CST model: 3.9% less $cv-R^2$ than CST pRFs with fixed temporal parameters and 1.5% less $cv-R^2$ than CST pRFs with optimized parameters. In particular, DN-ST pRFs tend to underpredict the level of simultaneous suppression for short stimulus timings (0.2 s) (**Supplementary Fig 9A**). The ANOVA also indicates a significant interaction between pRF model and ROI ($F(2,7)=5.0$, $p=1.6 \times 10^{-9}$), where both CST models perform significantly better than the DN-ST model in almost all visual areas, and the main CST pRF model explains slightly but significantly more $cv-R^2$ than the optimized CST pRF model in visual areas V1, hV4, V3A/B, LO1/2, and TO1/2 (**Supplementary Table 4**, post-hoc Bonferroni-corrected t -tests).”*

And [p. 22, lines 470-475]:

“Examining the relationship between simultaneous suppression and optimized CST model parameters

underscores our findings that larger pRF sizes, stronger compressive nonlinearity (i.e., smaller exponents), and contributions of both sustained and transient channels are important for predicting the level of simultaneous suppression across the visual hierarchy, whereas time constant parameters do not systematically co-vary with observed suppression levels (Supplementary Fig 9B-D)."

R2.4: One other question about the model is that the present CST model has a fixed power-law exponentiation analogous to the CSS model. However, with the Groen and earlier Zhou studies showing an explicit divisive normalization pool analogous to the explicit spatial normalization pool of the recent Aqil et al study, would a combination of these explicit temporal and spatial models not be a more appropriate modeling strategy? I don't mean to insinuate that the authors should perform this modeling, but I hope they could at least discuss this in more detail.

Response 2.4: We used a fixed static nonlinearity in both CSS and CST, as it allows for a closer comparison between the two models. When predicting the level of simultaneous suppression with the spatiotemporal delayed normalization pRF (DN-ST) model in the 7 subjects that have parameter estimates from Kim et al. 2024 we find overall no big difference in response dynamics (see example voxel in **Supplementary Fig 9A**) or cross-validated variance explained (**Supplementary Fig 8B,C**). One difference we noticed between the DN-ST and CST-optimized model is that the DN-ST model tends to predict less simultaneous suppression across the visual hierarchy for short stimulus durations than observed. In terms of pRF parameters, we find that for the DN-ST model an increase in suppression levels across the visual hierarchy correlates with larger pRF sizes, smaller exponents, smaller semi-saturation constants (reflecting more compression), and smaller tau_1 and tau_2 values (reflecting the time constant of the exponential decay) (**Supplementary Fig 9C**). This suggests that both types of compression implementation (static nonlinearity and delayed divisive normalization) in a spatiotemporal pRF model can predict suppression effects.

Action 2.4: We added a new analysis using a DN-ST model to predict simultaneous suppression. Results are shown in **Supplementary Figs 8-9 and Supplementary Table 4 [pp. 54-57]**, described on **p. 19** (first paragraph, lines 424-439), and **p. 21** (first paragraph, lines 475-480), and the model is described in Methods **[pp. 33-34]**.

R2.5: the explanatory figure 1 is somewhat confusing, and in my mind needs work. Mostly, much of the information is repeatedly depicted and it's unclear why that is. So as a reader, I end up looking for reasons why things are shown 2 or even 3 times - rather puzzling.

- only 2 conditions are shown in A, and then repeated in B, but for a reader it's unclear why these specific 2 conditions are repeated in B.
- In B, everything about the timings is displayed twice when the difference between the top and bottom subpanels is just spatial size.
- there is no visual indication how the 8s trials are embedded in the experimental run, but signal timecourses in figure 2 are from -5 to 15 s. I end up wondering about the blanks in between, something I have to look up in the methods.

Response 2.5: Thank you for this suggestion. It is difficult to visualize a 2x2x2 spatiotemporal design without repeating information, but we agree with the Reviewer that our choice of repeating some, but not other items may be confusing for the reader. We also agree that we can visualize information about chunking the time course more explicit into this figure. We have revised **Fig 1** to better illustrate the experimental design.

Action 2.5: Fig 1 [p. 7] now contains the entire time series of a single SEQ-SIM run, where we zoom into four different blocks: 2 SEQ and 2 SIM, both with long and short stimulus timings. We also created a 3D grid to convey the 2x2x2 design.

REVIEWERS' COMMENTS

Reviewer #1 (Remarks to the Author):

The authors have done an extremely thorough job in addressing all my points - I enjoyed reading the responses. I agree that the manuscript is now even stronger and I have only one minor comment which I think it will be easy to address: The authors model surround suppression as a DOG. But surround suppression is often thought of as a nonlinear contrast gain control computation (the surround contributes to the denominator of the normalization equation - effectively reducing the input contrast). I could not quite make out from the responses if this model had also been implemented as an alternative?

Reviewer #2 (Remarks to the Author):

I want to thank the authors for their thoughtful reply and revisions.

REVIEWER COMMENTS

We thank both reviewers for their positive response to our revised manuscript. We addressed the question by Reviewer #1 blue below. We hope that with these new changes, both Reviewers will accept the revised manuscript for publication.

Reviewer #1 (Remarks to the Author):

The authors have done an extremely thorough job in addressing all my points - I enjoyed reading the responses. I agree that the manuscript is now even stronger and I have only one minor comment which I think it will be easy to address: The authors model surround suppression as a DOG. But surround suppression is often thought of as a nonlinear contrast gain control computation (the surround contributes to the denominator of the normalization equation - effectively reducing the input contrast). I could not quite make out from the responses if this model had also been implemented as an alternative?

Thank you for these kind words in support of our revised manuscript!

Response: As the Reviewer notes, there are multiple ways of implementing surround suppression. We implemented the Difference of Gaussian (DoG) model published by Zuiderbaan *et al.* 2012 JoV. This model predicts the total pRF response by subtracting the response from the surround pRF from its center pRF response. We used the center-surround scale factors published by Aqil *et al.* 2021 PNAS to estimate the estimate the size of surround pRFs in the Zuiderbaan *et al.* 2012 JoV DoG model. We did not implement the divisive normalization center-surround model described by Aqil *et al.* We note that the model by Aqil *et al.* is also a spatial-only pRF model, like the Zuiderbaan *et al.* pRF model, and therefore does not predict the observed differences in simultaneous suppression between the two stimulus timing presentations (short: 0.2 s and long: 1 s).

Action: We have now clarified this detail in the Results section (p.19) and Methods section (p. 34-35) by only referring to Zuiderbaan *et al.* when first describing the DoG pRF model on. We now only refer to Aqil *et al.* when describing the average scale factor on p. 35 and clarified why we used Aqil *et al.* scale factors: *“Difference of Gaussians (DoG) pRF model. To test whether center-surround DoG pRFs predict simultaneous suppression, we simulate each voxel’s pRF as two 2D Gaussians, a center from which a larger surround is subtracted⁴⁸. ...*

*The center Gaussian is identical to the LSS pRF, estimated from the retinotopy session (Equation 2). The surround Gaussian has the same center position with a larger size, where the scale factor was based on the average center/surround size ratio from Aqil *et al.*⁵⁰; V1: 7.4, V2: 6.8, V3: 7.3, and hV4: 5.8 times the center size. We used a constant scaling for all voxels within the same visual area, because directly estimating DoG pRFs from the independent retinotopy data using the approach by Zuiderbaan *et al.*⁴⁸ resulted in unstable model fits. This instability is likely due to the relatively few and short blank periods in our retinotopy experiment compared to Zuiderbaan *et al.*, which has 4 x 30s-blank periods for each 5.5-min run. **We used scale factors by Aqil *et al.* as Zuiderbaan *et al.* does not report average center-surround scale factors within a visual area and data are limited to V1, V2, and V3. We did not implement the divisive normalization pRF model as described by Aqil *et al.*”***

Reviewer #2 (Remarks to the Author):

I want to thank the authors for their thoughtful reply and revisions.

Thank you for your feedback and supporting the publication of our manuscript!